Corrected: Author correction

# RNA cytosine methylation and methyltransferases mediate chromatin organization and 5-azacytidine response and resistance in leukaemia

Jason X. Cheng[1,2], Li Chen[1], Yuan Li[1,3], Adam Cloe[1], Ming Yue[1], Jiangbo Wei[4], Kenneth A. Watanabe[5], Jamile M. Shammo[6], John Anastasi[1], Qingxi J. Shen[7], Richard A. Larson[2,8], Chuan He [2,4], Michelle M. Le Beau[2,8] & James W. Vardiman[1]

The roles of RNA 5-methylcytosine (RNA:m5C) and RNA:m5C methyltransferases (RCMTs) in lineage-associated chromatin organization and drug response/resistance are unclear. Here we demonstrate that the RCMTs, namely NSUN3 and DNMT2, directly bind hnRNPK, a conserved RNA-binding protein. hnRNPK interacts with the lineage-determining transcription factors (TFs), GATA1 and SPI1/PU.1, and with CDK9/P-TEFb to recruit RNA-polymerase-II at nascent RNA, leading to formation of 5-Azacitidine (5-AZA)-sensitive chromatin structure. In contrast, NSUN1 binds BRD4 and RNA-polymerase-II to form an active chromatin structure that is insensitive to 5-AZA, but hypersensitive to the BRD4 inhibitor JQ1 and to the downregulation of NSUN1 by siRNAs. Both 5-AZA-resistant leukaemia cell lines and clinically 5-AZA-resistant myelodysplastic syndrome and acute myeloid leukaemia specimens have a significant increase in RNA:m5C and NSUN1-/BRD4-associated active chromatin. This study reveals novel RNA:m5C/RCMT-mediated chromatin structures that modulate 5-AZA response/resistance in leukaemia cells, and hence provides a new insight into treatment of leukaemia.

[1] Department of Pathology, University of Chicago, Chicago, IL 60637, USA. [2] University of Chicago Comprehensive Cancer Center, Chicago, IL 60637, USA. [3] Department of Haematology, Peking University First Hospital, Beijing 100034, China. [4] Department of Chemistry, University of Chicago, Chicago, IL 60637, USA. [5] Genomics Core, Emory University, Atlanta, GA 30322, USA. [6] Rush University Medical Center, Chicago, IL 60612, USA. [7] University of Nevada, Las Vegas, NV 89154, USA. [8] Department of Medicine, University of Chicago, Chicago, IL 60637, USA. These authors contributed equally: Li Chen, Yuan Li. Correspondence and requests for materials should be addressed to J.X.C. (email: Jason.Cheng@uchospitals.edu)

A large number of RNA modifications have been identified in the past[1], but the role of RNA modifications and their modifying enzymes, i.e. writers, readers and erasers, in gene regulation and chromatin organization remain largely unexplored[2, 3]. To date, the published studies have been largely focused on RNA N[6]-methyladenosine (RNA:m[6]A) and its modifying enzymes, and little attention has been paid to RNA 5-methylcytosine (RNA:m[5]C) and its modifying enzymes[3]. Currently, 57 RNA methyltransferases have been identified in humans[4]. At least ten are RNA:m[5]C methyltransferases (RCMTs), including NSUN1 to NSUN7, NSUN5a/b/c, and DNMT2. NSUN2/Misu was first identified in yeast as multisite-specific tRNA:m[5]C methyltransferase 4 (Trm4)[5, 6]. The human homologue of yeast Trm4, namely TRM4, MISU or NSUN2, can methylate 5-cytosine in various non-coding RNAs[7, 8] and plays an important role in the regulation of stem cell development and cancer cell proliferation and metastasis[9]. DNMT2 was originally considered to be a DNA methyltransferase, but now is recognized as an RNA/tRNA methyltransferase[10]. DNMT2 has been shown to play an important role in organ development and stress-induced tRNA cleavage[11]. It is upregulated in hundreds of tumour samples in the COSMIC database, and more than 60 somatic mutations in *DNMT2* have been identified in tumours of various tissue types[12, 13]. NSUN2 and DNMT2 are also involved in the regulation of responses to 5-fluorouracil and 5-AZA, respectively[14, 15]. *NSUN1* encodes a proliferation-associated nucleolar protein known as NOL1 or NOP2[16, 17]. A pathogenic *NSUN1/NOL1-E2A* fusion has been identified in rare B-lymphoblastic leukaemia cases[18]. A recent study demonstrated that NSUN1/NOL1 interacts with telomerase to regulate cyclin D1 expression[19]. RNA-binding proteins (RBPs) are also known to play a pivotal role in gene regulation and chromatin organization[20–22]. Among them hnRNPK is an evolutionarily conserved member of the heterogeneous nuclear ribonucleoprotein (hnRNP) family that can bind pre-mRNA and impact mRNA splicing, export and translation[23]. hnRNPK binds preferentially and tenaciously to poly(C) via three repeats of a motif, termed K-homology domain[24]. Aberrantly elevated hnRNPK levels have been linked to various forms of cancer, including myeloid neoplasms[25]. Elevated levels of hnRNPK are also correlated with the levels of the BCR-ABL1 fusion proteins and disease progression in chronic myeloid leukaemia (CML)[26, 27]. Paradoxically, *Hnrnpk* haploinsufficiency in mice leads to an increase in the development of myeloid leukaemia[28]. HnRNPK is required for P53-dependent anticancer therapy[29–31]. Azacitidine (5-AZA), a DNA hypomethylating agent, is widely used to treat various haematologic malignancies, such as myelodysplastic syndrome (MDS) and acute myeloid leukaemia (AML). Although a vast majority (~90%) of 5-AZA is incorporated into RNA[32], it is still unknown whether RNA:m[5]C, RCMTs and hnRNPK play a role in the response/resistance to 5-AZA in leukaemia cells.

Here we demonstrate that RCMTs interact with different partners to form distinct complexes and active chromatin structures at nascent RNA in 5-AZA-sensitive leukaemia cells (ASLCs) vs. 5-AZA-resistant leukaemia cells (ARLCs). Such chromatin structures are important for differential response/resistance to 5-AZA and survival of the leukaemia cells. Based on our data, we propose a working model in which distinct RNA:m[5]C/RCMT-mediated chromatin structures are formed in ASLCs vs. the ARLCs. A significant increase in RNA:m[5]C and NSUN1-/BRD4-associated active chromatin is observed in clinical 5-AZA-resistant MDS/AML specimens, supporting the importance and clinical relevance of our working model.

## Results

### Increased levels of RNA:m[5]C, RCMTs and hnRNPK in ARLCs.
To identify the factors that affect the response and resistance to epigenetic drugs in leukaemia, we performed two sets of initial experiments, testing multiple epigenetic drugs in various myeloid leukaemia cell lines with different lineages, growth rates and cytogenetic abnormalities and establishing multiple drug-resistant leukaemia cell lines with various nucleic acid analogues/drugs. For establishing drug-resistant cell lines, five nucleic acid analogues, 5-AZA, decitabine (DC), fluorocytidine (5-FC), 5-fluorodeoxycytidine (5-FDOC) and gemcitabine (GC), were used. Our data demonstrated a correlation between drug resistance and the deoxyribose or ribose-phosphate chemical backbones of the nucleic acid analogues (Supplementary Fig. 1). For example, the RNA-based drugs, 5-AZA and 5-FC, shared a similar drug response in the 5-AZA-sensitive and 5-AZA-resistant erythroid leukaemia cell lines, OCI-M2 and M2AR (Supplementary Fig. 1b), as well as in the 5-AZA-sensitive and 5-AZA-resistant monocytic leukaemia cell lines, SC and SCAR (Supplementary Fig. 1c). In contrast, the DNA-based drugs, DC and 5-FDOC, had a different drug response in these leukaemia cell lines. These results led us to explore the possibility of an RNA-mediated mechanism for 5-AZA action and response/resistance. Western blot analyses demonstrated a mild increase (~3 fold) in RCMTs, including NSUN1, NSUN3, DNMT2 and hnRNPK, in the 5-AZA-resistant erythroid and monocytic leukaemia cell lines, M2AR and SCAR, as compared to the corresponding, originally 5-AZA-sensitive OCI-M2 and SC leukaemia cell lines, respectively (Fig. 1a). Dot blotting with anti-5-methylcytosine (5-mC, m[5]C) antibody demonstrated markedly increased m[5]C in RNA but not in DNA in the 5-AZA-resistant M2AR and SCAR leukaemia cells compared to the 5-AZA-sensitive OCI-M2 and SC leukaemia cells (Fig. 1b, c). Consistent with the dot blot results, crosslink-assisted DNA modification immunoprecipitation assay showed that 5-AZA treatment (at 1 µM for 3 h) did not change global DNA 5-methylcytosine (DNA:m[5]C) and 5-hydroxyomethylcytosine (DNA:hm[5]C) levels (Supplementary Fig. 2b). In contrast, 5-AZA induced marked changes in DNA:m[5]C and DNA:hm[5]C at the regulatory region of *SPI1/PU.1*, a key myeloid-determining gene, in OCI-M2 and SC cells (Supplementary Fig. 2c,d,e). We developed an agarose bead assay to visualize and quantitate the binding of hnRNPK to RNA using purified hnRNPK protein (Fig. 1d). The experiments showed purified recombinant hnRNPK preferentially bound the cytosine-methylated RNA oligo over the unmethylated RNA oligo at a ratio of methylated poly-(C)/unmethylated poly-(C) of ~20 fold (Fig. 1d, e). Binding of the endogenous hnRNPK to unmethylated and cytosine-methylated RNA was also demonstrated using nuclear lysate from human leukaemia cells. The endogenous hnRNPK from both OCI-M2 and M2AR cells bound to the cytosine-methylated RNA stronger (~5 fold) than to unmethylated RNA (Supplementary Fig. 3). The endogenous hnRNPK from the 5-AZA-resistant M2AR cells had higher (~10 fold) binding to both the unmethylated and cytosine-methylated RNA oligos than that from the 5-AZA-sensitive OCI-M2 cells (Supplementary Fig. 4). These data demonstrate that compared to ASLCs, ARLCs had increased levels of RNA:m[5]C, RCMTs and hnRNPK with preferential binding to cytosine-methylated RNA, suggesting that RNA:m[5]C, RCMTs and hnRNPK may be involved in 5-AZA response/resistance in leukaemia cells.

### Distinct functional RCMT complexes in leukaemia cells.
RNase digestion-coupled immunoprecipitation (IP) and co-immunoprecipitation (co-IP) demonstrated RNA-independent, direct binding of hnRNPK to a subset of RCTMs, namely, DNMT2 and NSUN3, in OCI-M2 leukaemia cells (Fig. 2a). In contrast, no appreciable interactions between hnRNPK and NSUN1 or NSUN2 were detected in OCI-M2 leukaemia cells

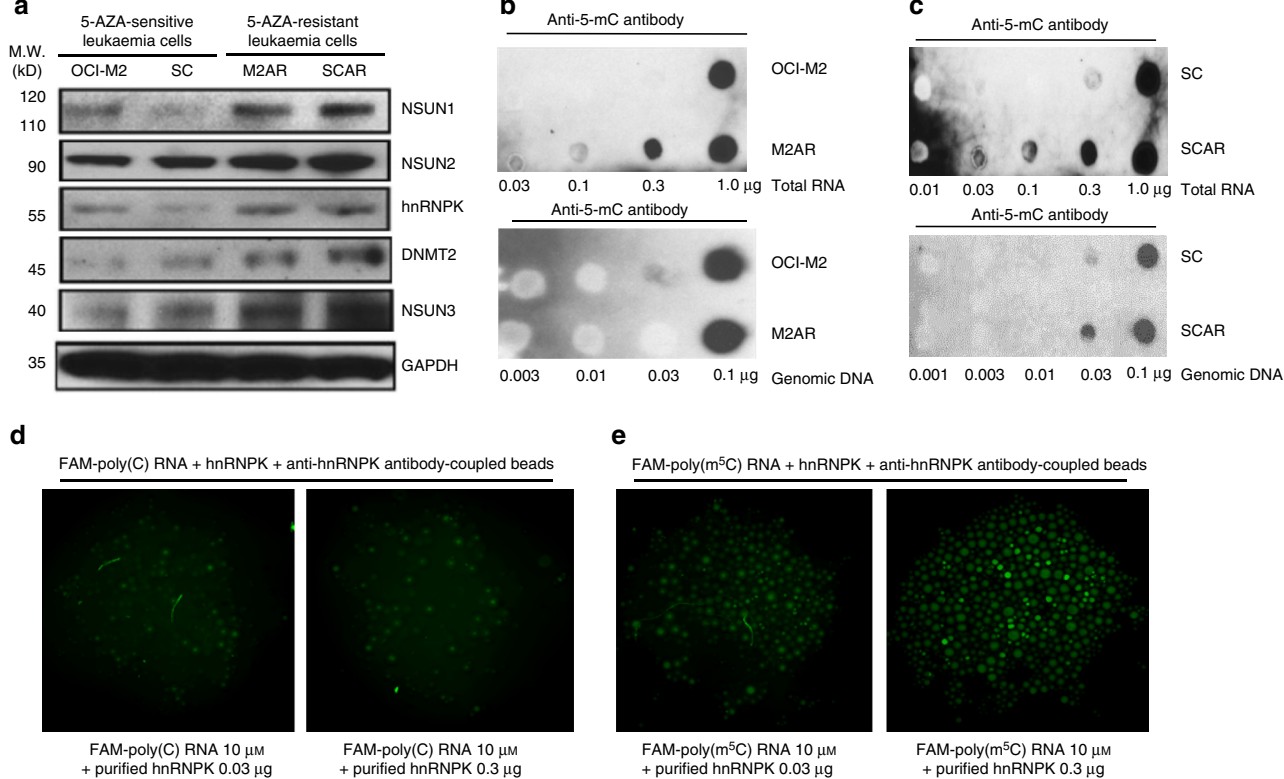

**Fig. 1** Differential expression of RNA:m5C, RCMTs and hnRNPK in 5-AZA-sensitive and 5-AZA-resistant leukaemia cells and the binding of hnRNPK to unmethylated and cytosine-methylated RNA. **a** Western blot analysis of expression of RCMTs, hnRNPK and other proteins in the 5-AZA-sensitive OCI-M2 and SC leukaemia cells and the 5-AZA-resistant M2AR and SCAR leukaemia cells. **b** Dot blot analysis of 5-methylcytosine (m5C) in RNA and DNA from OCI-M2 and M2AR cells. **c** Dot blot analysis of 5-methylcytosine (m5C) in RNA and DNA from SC and SCAR cells. **d**, **e** Visualization and measurement of the binding of purified recombinant hnRNPK to the unmethylated and cytosine-methylated fluorescein (FAM)-labelled RNA oligos by an antibody-coupled bead assay

(Fig. 2a). To further determine the functions of hnRNPK and RCMTs, a series of siRNA knockdown experiments were performed to knockdown hnRNPK and individual RCMTs. As shown in Fig. 2b, knockdown of hnRNPK, DNMT2 or NSUN3 caused a loss of not only the targeted proteins but three other proteins as well, suggesting that hnRNPK, DNMT2 and NSUN3 form a functional complex. Knockdown of hnRNPK also slightly reduced SPI1/PU.1 but had no significant effects on total RNA polymerase II (RNA-pol-II). In contrast, siRNA knockdown of NSUN1 or NSUN2 significantly reduced both the NSUN1 and NSUN2 proteins, but had little effects on the hnRNPK/NSUN3/ DNTM2 complex and total RNA-pol-II in the leukaemia cells (Fig. 2c). Knockdown of hnRNPK or any of the RCMTs significantly inhibited the growth of SC leukaemia cells (Fig. 2d). Knockdown experiments using the same sets of siRNAs were performed in OCI-M2 cells and showed similar, but less dramatic effects, than those in SC cells due to a much lower transfection efficiency in OCI-M2 than that in SC. The transfection efficiencies for OCI-M2 cells and SC cells were ~20 and ~70%, respectively. Together, these data suggest selective interactions between hnRNPK and various RCMTs to form distinct functional complexes important for integrity of these proteins and survival of the leukaemia cells.

**5-AZA induces cell lineage-associated chromatin changes.** We performed the epigenetic drug screening in multiple myeloid leukaemia cell lines in our initial experiments (Supplementary Table 1). We observed distinct patterns of lineage-associated response to the epigenetic drugs. To our surprise, the drug

responsive patterns were not affected by the cytogenetic abnormalities and proliferation rates of the leukaemia cell lines (Supplementary Table 1 and Supplementary Fig. 3). Simulated emission depletion (STED) confocal microscopy was performed to elucidate subcellular localization and co-localization of the lineage-determining transcription factors (TFs), GATA1 and SPI1/PU.1, and the methylcytosine dioxygenase, TET2, and their associated chromatin structural changes in response to 5-AZA (Fig. 3a, b). GATA1 and TET2 are (partially) co-localized and formed dot-like chromatin structural patterns, and 5-AZA caused disruption of GATA1-/TET2-associated chromatin structure in OCI-M2 cells (Fig. 3a). Similar patterns of subcellular localization and drug-responsive chromatin structural changes of SPI1/PU.1 and TET2 were observed in untreated and 5-AZA-treated SC cells (Fig. 3b). The erythroid cell line OCI-M2 and the monocytic cell line SC had different lineage-associated responses to epigenetic drugs (Fig. 3c, Supplementary Table 1 and Supplementary Fig. 5). Chromosomal Conformation Capture (3C) and Chromatin ImmunoPrecipitation (ChIP) with an antibody against active RNA-pol-II (CTD-S2P/S5P) were performed to elucidate the effects of 5-AZA on the chromatin conformation at the SPI1/PU.1 locus in two AML cell lines with an erythroid immunophenotype, OCI-M2 derived from high-grade MDS and K562 derived from an erythroid blast crisis of CML, and two AML cell lines with a monocytic immunophenotype, SC and THP1 that were derived from the peripheral blood monocytic leukaemia cells. The primer pairs for real-time PCRs are located between the proximal promoter and the distal enhancer of SPI1/PU.1 (Fig. 3d). The positive control using a closely located primer pair (1L and 1R) at the promoter produced the strongest signals in the leukaemia cells

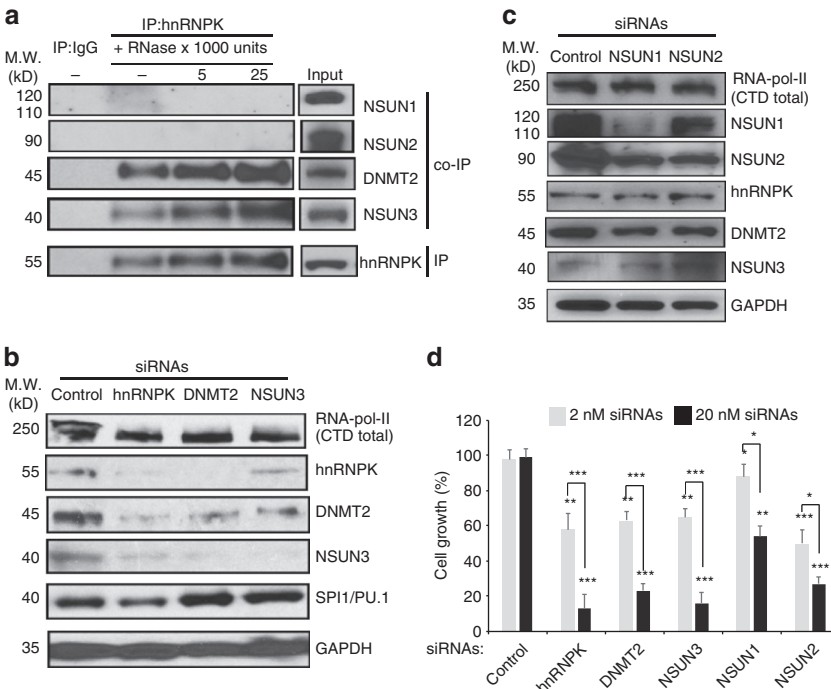

**Fig. 2** Distinct functional hnRNPK/RCMT complexes required for leukaemia cell survival. **a** RNase digestion-coupled immunoprecipitation (IP) and co-immunoprecipitation (co-IP) to examine the interactions between hnRNPK and RCMTs, including DNMT2, NSUN3, NSUN1 and NSUN2, in OCI-M2 cells. **b** Measurements of the expression of hnRNPK, DNMT2, NSUN3 and other proteins in SC leukaemia cells treated with control siRNAs and the siRNAs targeting HNRNPK, DNMT2 and NSUN3 at day 3. **c** Measurements of the expression of NSUN1, NSUN2 and other proteins in SC leukaemia cells treated with control siRNAs and the siRNAs targeting NSUN1 and NSUN2 at day 3. **d** Measurements of the growth of SC leukaemia cells treated with control siRNAs and siRNAs targeting hnRNPK and individual RCMTs by MTT assay at day 3. Data are presented as the mean ± SEM of $n = 3$ independent samples. *$P < 0.05$, **$P < 0.01$ and ***$P < 0.001$ by Student's $t$ test

(Fig. 3e, f). The negative and the background controls were done with the 1L + 2L and the 1L + 3R primer pair, respectively. Only a background level of PCR signals with the primer pairs (1L + 4R) located in the *SPI1/PU.1* promoter and enhancer was detected in the untreated erythroid leukaemia cells, indicating that no promoter-enhancer chromatin loop was formed in these cells (Fig. 3e, left). In contrast, 5-AZA markedly increased the PCR signals (~12 fold) with the same pair of primers (1L + 4R), suggesting the presence of a 5-AZA-induced chromatin loop between the promoter and the enhancer (Fig. 3e, right). Strong PCR signals were detected with the same pair of primers in the untreated monocytic leukaemia cells, SC and THP1, indicating an existing promoter-enhancer loop in these cells (Fig. 3e, right). In the 5-AZA-treated monocytic leukaemia cells, the PCR signals were markedly reduced, indicating disruption of the promoter-enhancer loop (Fig. 3e, right). ChIP with an antibody against active RNA-pol-II (RNA-pol-II CTD-S2/S5P) detected a very low level of active RNA-pol-II recruitment at the *SPI1/PU.1* promoter and enhancer in the untreated OCI-M2 and K562 cells (Fig. 3f, left). 5-AZA significantly increased (approximately six fold) active RNA-pol-II recruitment to the *SPI1/PU.1* promoter and enhancer in OCI-M2 and K562 cells (Fig. 3f, left). In contrast, the same 5-AZA treatment significantly decreased active RNA-pol-II recruitment to these regions in SC and THP1 cells (Fig. 3f, right). Figure 3g schematically summarizes the data suggesting that 5-AZA induces lineage-associated, locus-specific chromatin structural changes in leukaemia cells.

**RNA/hnRNPK mediates TF−chromatin modifier interactions**. To explore the mechanisms underlying the lineage-associated 5-AZA-responsive chromatin structural changes, we performed western blotting, IP and co-IP to examine the expression and the

interactions of the lineage-determining TFs, GATA1 and SPI1/PU.1, and various chromatin modifiers. As shown in Fig. 4, chromatin modifiers, including TET1,2,3, DNMT1, DNMT3A/3B and EZH1/2, were expressed at more or less the same levels in both OCI-M2 and SC cells (Fig. 4a, input). As expected, OCI-M2 cells had a high level of GATA1 but a very low level of SPI1/PU.1, and SC cells expressed only SPI1/PU.1 but not GATA1 (Fig. 4a, input). In OCI-M2 cells GATA1 specifically interacted with TET2 and DNMT3A/3B, whereas SPI1/PU.1 specifically interacted with DNMT1 and EZH2 (Fig. 4a, left). It is worth noting that GATA1 also interacted with SPI1/PU.1 in our co-IP studies; such a direct GATA1-SPI1/PU.1 interaction has been reported previously in other erythroid leukaemia cell lines[33–35]. In contrast, SPI1/PU.1 selectively interacted with TET2, but not DNMT1 or EZH2 (Fig. 4a, right). The interactions between the lineage-determining factors and DNA modifiers were highly sensitive to 5-AZA (Fig. 4b, c). 5-AZA rapidly disrupted the interaction between SPI1/PU.1 and DNMT1 in OCI-M2 (Fig. 4b) as well as the interaction between the SPI1/PU.1 and TET2 in SC (Fig. 4c). RNase digestion-coupled IP and co-IP demonstrated a dose-dependent RNase-induced disruption of the GATA1−TET2 interaction and the SPI1/PU.1−DNMT1 interaction in OCI-M2 cells, whereas the SPI1/PU.1−EZH2 interaction was unaffected by RNase (Fig. 4d). The interaction between SPI1/PU.1 and TET2 interaction in SC cells was also highly sensitive to RNase-digestion (Fig. 4e), suggesting that RNA mediates the interaction between TET2 and GATA1 or SPI1/PU.1. However, we could not detect appreciable RNA-binding by TET2 or the lineage-determining factors (GATA1 and SPI1/PU.1) at physiological concentrations of these proteins. Therefore, we screened a small pool of commercially available antibodies against hnRNP family proteins using antibodies against GATA1 (in OCI-M2) and SPI1/

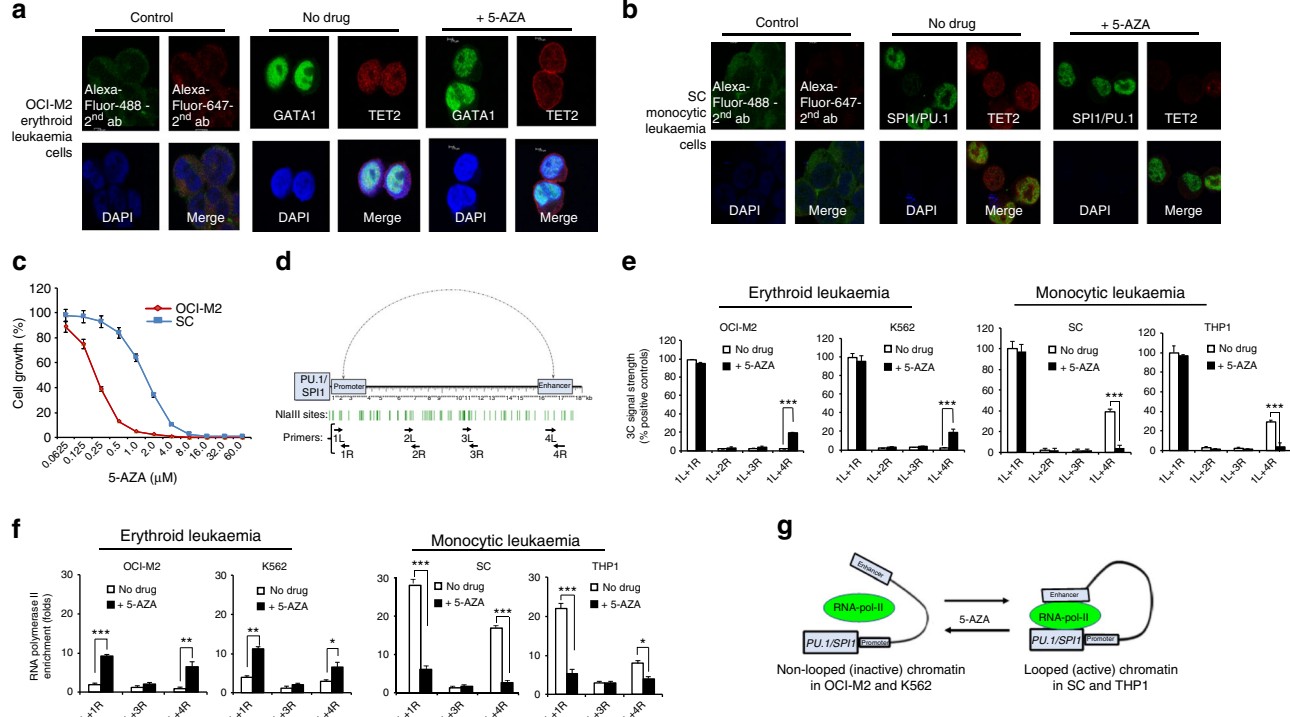

**Fig. 3** Lineage-associated drug-responsive chromatin structural changes in leukaemia cells. **a** STED confocal microscopy analysis of the subcellular localization and co-localization of GATA1 and TET2 in OCI-M2 cells with no drug and 1 μM 5-AZA for 3 h. **b** STED confocal microscopy analysis of the subcellular localization and co-localization of GATA1 and TET2 in SC cells with no drug and 1 μM 5-AZA for 3 h. **c** Differential growth inhibition of OCI-M2 and SC leukaemia cells induced by 1 μM 5-AZA for 3 days. **d** A schematic illustration of the locations of the PCR primers designed in the regulatory region of *SPI1/PU.1*. **e** The PCR signals detected by 3C assays in the erythroid leukaemia cell lines, OCI-M2 and K562, and the monocytic leukaemia cell lines, SC and THP1, with no drug and 1 μM 5-AZA for 3 h. **f** ChIP assays with an antibody against the active form (CTD-S2P/S5P) of RNA-pol-II at the *SPI1/PU.1* locus in the OCI-M2, K562, SC and THP1 cells, with no drug and 1 μM 5-AZA for 3 h. **g** A schematic summary of these data suggesting opposite transformations of the chromatin conformation at *SPI1/PU.1* in erythroid vs. monocytic leukaemia cells in response to 5-AZA. Data are presented as the mean ± SEM of $n = 3$ independent samples. $*P < 0.05$, $**P < 0.01$ and $***P < 0.001$ by Student's *t* test

PU.1 (in SC) as IP "baits" to identify the interacting RNP(s). We identified hnRNPK as a unique RBP that directly interacted with both GATA1 and TET2, whereas other RNPs, such as hnRNPA1 and AUF1, did not bind GATA1 (Fig. 4f). A similar pattern of the interactions between hnRNPK and SPI1/PU.1 as well as TET2 was observed in SC cells (Supplementary Fig. 6). Together, these data demonstrate that RNA/hnRNPK mediates 5-AZA-sensitive interactions between the lineage-determining TFs and chromatin modifiers in leukaemia cells as schematically illustrated in Fig. 4g.

**hnRNPK binds CDK9 to form 5-AZA-sensitive active chromatin.** CDK7/TFIIH and CDK9/P-TEFb can phosphorylate the serine 5 and serine 2 residues in the C-terminal domain (CTD) of RNA-pol-II, i.e. CTD-S5P and CTD-S2P, which are critical for transcription initiation and elongation, respectively[24, 36]. Western blotting showed that 5-AZA significantly reduced CTD-S2P in OCI-M2 cells within 1.5 h while levels of total RNA-pol-II, CTD-S5P, RCMTs, hnRNPK CDK9/P-TEFb and CDK7 remained unchanged (Fig. 5a). RNase digestion-coupled IP and co-IP revealed a direct interaction between hnRNPK and CDK9/P-TEFb and indirect interactions between hnRNPK and CDK7/TFIIH as well as RNA-pol-II CTD-S2P in the OCI-M2 leukaemia cells (Fig. 5b). By employing a newly developed 5-ethynyluridine (EU) clicking chemistry technology[37] and STED confocal microscopy, we demonstrated co-localization of hnRNPK and active RNA pol-II (CTD-S2P) at nascent RNA, and the co-localization was markedly increased in the 5-AZA-resistant M2AR cells compared to the 5-AZA-sensitive OCI-M2

leukaemia cells (Fig. 5c). 5-AZA rapidly dissociated hnRNPK from the RNA-pol-II/CDK9/7 complex within 10–20 min, while the expression of these proteins remained unchanged (Fig. 5d). In contrast, the interactions between hnRNPK and DNMT2 as well as NSUN3 were insensitive to 5-AZA in the same leukaemia cells (Fig. 5e). Based on all the above-described data, we hypothesized a working model of the hnRNPK/RCMT-mediated lineage-associated, 5-AZA-sensitive active chromatin structure at nascent RNA in leukaemia cells and a novel 5-AZA action mechanism through disruption of the hnRNPK-associated active chromatin structure (Fig. 5f).

**NSUN1/BRD4 directly bind RNA-pol-II CTD-S2P in ARLCs.** To explore the role of the RNA-pol-II and RCMT complexes in 5-AZA resistance, we compared the expression levels of the components of these complexes. Western blotting demonstrated a small increase (~1–2 fold) in the expression of total RNA-pol-II, CTD-S2P, CDK7, CDK9/P-TEFb, BRD4 and BRD2 in the 5-AZA-resistant M2AR and SCAR leukaemia cells compared to the original 5-AZA-sensitive OCI-M2 and SC leukaemia cells (Fig. 6a). As expected, the 5-AZA-resistant M2AR leukaemia cells had markedly increased (~20 fold) hnRNPK-associated RNA-pol-II CTD-S2P compared to ASLCs (Fig. 6b). To our surprise, the markedly increased interactions between hnRNPK and CTD-S2P as well as GATA1 in the 5-AZA-resistant M2AR leukaemia cells were still highly sensitive to 5-AZA, and 5-AZA rapidly disrupted these interactions within 10 min in these cells (Fig. 6b), suggesting an hnRNPK-independent mechanism for 5-AZA-resistance.

These data led us to search for the factors that mediate the interactions between RCMTs and RNA-pol-II CTD-S2P in the ARLCs. Reverse IP with anti-RNA-pol-II CTD-S2P antibody and co-IPs with antibodies against various RCMTs identified NSUN1 as a unique RCMT that bound RNA-pol-II CTD-S2P in the 5-AZA-resistant M2AR leukaemia cells (Fig. 6c). The RNA-pol-II CTD−S2P−NSUN1 interaction was present only in the 5-AZA-

resistant cells but not in the 5-AZA-sensitive cells (Fig. 6c). The interactions between RNA-pol-II CTD-S2P and NSUN1 as well as RNA-pol-II CTD-S5P were resistant to 5-AZA in the 5-AZA-resistant M2AR leukaemia cells (Fig. 6c). In contrast, the markedly increased interactions between RNA-pol-II CTD-S2P and NSUN3 as well as DNMT2 in these 5-AZA-resistant M2AR leukaemia cells were highly sensitive to 5-AZA (Fig. 6c). IPs with

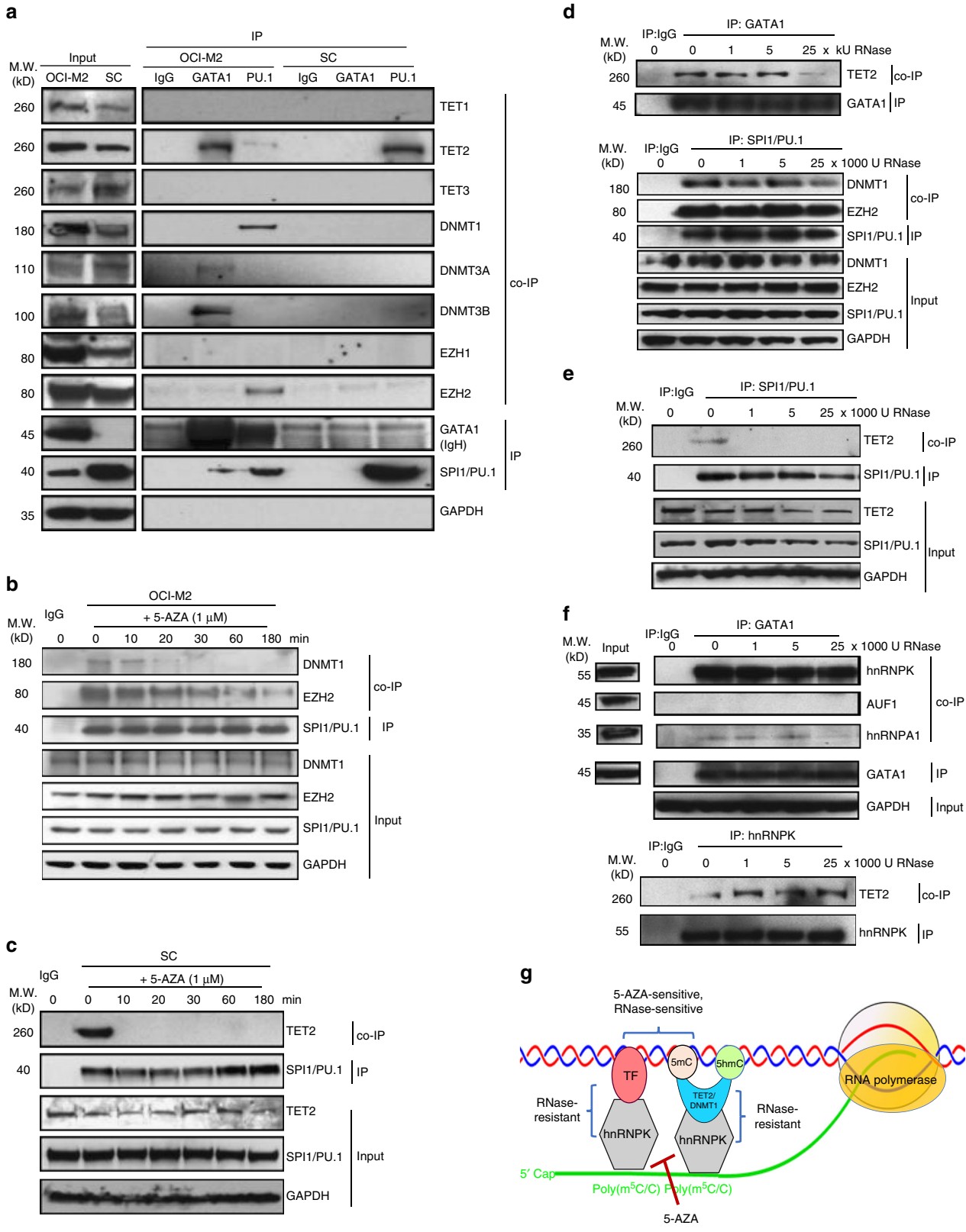

anti-BRD4 antibody and co-IP with antibodies against various components of the active RNA-pol-II and RCMT complexes demonstrated an ~80-fold increase in BRD4-associated RNA-pol-II CTD-S2P in the 5-AZA-resistant M2AR leukaemia cells compared to the 5-AZA-sensitive OCI-M2 cells (Fig. 6d), but no increase in the BRD4-bound CDK9/P-TEFb was observed in the 5-AZA-resistant M2AR cells. NSUN1 specifically interacted with BRD4 in the 5-AZA-resistant M2AR leukaemia cells (Fig. 6d). JQ1 is a thienotriazolodiazepine compound that can selectively disrupt the binding of BRD4 to acetylated lysine residues of histones in chromatin and inhibit BRD4-mediated transcription activation[38]. However, our data showed that the interactions between BRD4 and active RNA-pol-II CTD-S2P as well as NSUN1 were insensitive to JQ1. It is worth mentioning that the JQ1 induced a transient increase in the binding of BRD4 to active RNA-pol-II (CTD-S2P) and NSUN1 likely due to more free BRD4 released from the binding to acetylated lysine residues in chromatin by JQ1. The 5-AZA-resistant SCAR leukaemia cells had a similar pattern of the interactions between NSUN1 and RNA-pol-II CTD-S2P as well as BRD4 as seen in the above M2AR leukaemia cells (Supplementary Figs. 7 and 8, respectively). The above data reveal a unique NSUN1/BRD4/RNA-pol-II CTD-S2P complex in the ARLCs, suggesting a novel NSUN1-/BRD4-mediated mechanism for 5-AZA resistance in leukaemia cells.

**Distinct NSUN1/BRD4-associated active chromatin in ARLC.**
EU-clicking chemistry STED confocal microscopy illustrated the subcellular localization of NSUN1, BRD4 and RNA-pol-II CTD-S2P and their relationship to nascent RNA in M2 and M2AR cells (Fig. 7a, b). OCI-M2 cells had a weaker and more diffused pattern of NSUN1 and BRD4 (Fig. 7a, top). In contrast, M2AR cells showed stronger expression of NSUN1 and BRD4, and they seemed co-localized to form distinct nuclear peripheral zone-intensified granular particles at nascent RNA in M2AR cells (Fig. 7a, bottom). Similar patterns of subcellular localization and co-localization of NSUN1, RNA-pol-II CTD-S2P and nascent RNA were observed in OCI-M2 and M2AR cells as well (Fig. 7b). In situ proximity ligation rolling circle amplification (PL-RCA)[39, 40] was performed to further confirm the co-localization of NSUN1 with BRD4 and RNA-pol-II CTD-S2P. This experiment demonstrated a marked increase in the PL-RCA signals with the pairs of antibodies against NSUN1 and BRD4 (Fig. 7c) as well as NSUN1 and RNA-pol-II CTD-S2P in M2AR cells (Fig. 7d), compared to those in OCI-M2 cells. A similar pattern of the PL-RCA signals of co-localization of NSUN1 with BRD4 and RNA-pol-II CTD-S2P was observed with the same pairs of antibodies in SCAR vs. SC cells (Supplementary Figs. 9 and 10, respectively). These data demonstrate distinct NSUN1-/BRD4-associated active chromatin structure at nascent RNA in the ARLCs.

**Preferential inhibition of ARLCs by targeting NSUN1/BRD4.**
Based on our data, we hypothesized that ASLCs and ARLCs would have differential responses to inhibitors and siRNAs that target different RNA-pol-II CTD kinases and RCMTs. BS-181

and PHA-767491 are the selective inhibitors for the canonical RNA-pol-II CTD kinases, CDK7 and CDK9/P-TEFb, respectively[41, 42]. Experiments with BS-181 and PHA-767491 showed a preferential growth inhibition towards the ASLCs with a significantly higher growth inhibition on OCI-M2 than on M2AR (Fig. 8a). A similar pattern of growth inhibition with BS-181 and PHA-767491 was observed in SCAR vs. SC cells (Supplementary Fig. 11). JQ1 preferentially inhibited the growth inhibition of the 5-AZA-resistant leukaemia M2AR cells (Fig. 8b, top). There was marked synergistic growth inhibition between JQ1 and 5-AZA, and JQ1 re-sensitized the 5-AZA-resistant M2AR leukaemia cells to 5-AZA, resulting in growth inhibition of the leukaemia cells at very low concentrations of 5-AZA (Fig. 8b, bottom). Knockdown experiments with the siRNAs targeting various RCMTs demonstrated that downregulation of hnRNPK, NSUN3 and DNMT2 preferentially inhibited the growth of the 5-AZA-sensitive SC leukaemia cells (Fig. 8c, top). In contrast, knockdown of NSUN1 by siRNAs preferentially inhibited the 5-AZA-resistant SCAR leukaemia cells and re-sensitized SCAR cells to low concentrations of 5-AZA (Fig. 8c, bottom). Based on all the above-mentioned data, we developed a working model of distinctly different, drug-responsive active chromatin structures in ASLCs vs. ARLCs (Fig. 8d).

**Increased mRNA:m$^5$C in 5-AZA-resistant clinical AML/MDS cells.** To further determine the clinical significance of mRNA:m$^5$C in mediating 5-AZA resistance in AML and MDS, we performed a comparison study of mRNA:m5C levels by mass spectrometric analysis. Eighteen clinical bone marrow specimens were selected for this study, including nine initial bone marrows from 5-AZA-sensitive AML/MDS cases and nine bone marrows from 5-AZA-resistant (refractory) AML/MDS cases. These cases had various blast counts, immunophenotypes, cytogenetic abnormalities, and molecular alterations (Supplementary Table 2). A majority of these cases (11 of 18) showed monocytic differentiation with expression of CD11b, a monocytic lineage-associated marker, and lacked CD34 expression in the leukaemic blasts, and one of these cases had erythroid differentiation with expression of CD235a (glycophorin A), an erythroid lineage-associated antigen (Supplementary Table 3). As shown in Fig. 9a, mass spectra detected clear mRMA:m$^5$C and mRNA: adenosine (A) peaks and increased ratios of the m$^5$C/A in the bone marrow cells of two representative 5-AZA-resistant AML cases, compared to those in the bone marrow cells of two 5-AZA-sensitive AML cases. Quantitative analysis of mRNA:m$^5$C/A in these clinical bone marrow specimens demonstrated a statistically significant increase in mRNA:m$^5$C in the 5-AZA-resistant bone marrows compared to 5-AZA-sensitive bone marrows (Fig. 9b). Since the blast count varied greatly among these clinical specimens, we normalized the mRNA:m$^5$C/A values with manual blast counts in those cases. The blast count normalized mRNA:m$^5$C/A analysis demonstrated a very significant difference in mRNA: m$^5$C/A between the 5-AZA-resistant bone marrows and the 5-AZA-sensitive bone marrows (Fig. 9c). These mass spectrometric

**Fig. 4** RNA/hnRNPK-mediated selective interactions between lineage-determining TFs and chromatin modifiers. **a** Western blotting, IP with GATA1 and SPI1/PU.1 antibodies and co-IP with antibodies against various DNA and histone modifiers in OCI-M2 and SC cells. **b** Examination of the effects of 5-AZA on the interactions between SPI1/PU.1 and DNMT1 as well as EZH2 by co-IP in OCI-M2 cells treated with 1 μM 5-AZA at various time points. **c** Examination of the effects of 5-AZA on the interaction between SPI1/PU.1 and TET2 by co-IP in SC cells treated with 1 μM 5-AZA at various time points. **d** Examination of the RNA-dependence of the interaction between GATA1 and TET2 and the interactions between SPI1/PU.1 and DNMT1 as well as EZH2 by RNase digestion-coupled co-IP in OCI-M2 cells. **e** Examination of the RNA-dependence of the interaction between SPI1/PU.1 and TET2 by RNase digestion-coupled co-IP in SC cells. **f** Identification of hnRNPK as a direct binder of both GATA1 and TET2 by IP with GATA1 antibody and co-IPs with antibodies against various RNA-binding proteins, such as hnRNPA1 and AUF1, in OCI-M2 cells. **g** A schematic illustration of the nature of the interactions between hnRNPK and GATA1 as well as TET2 in OCI-M2 cells

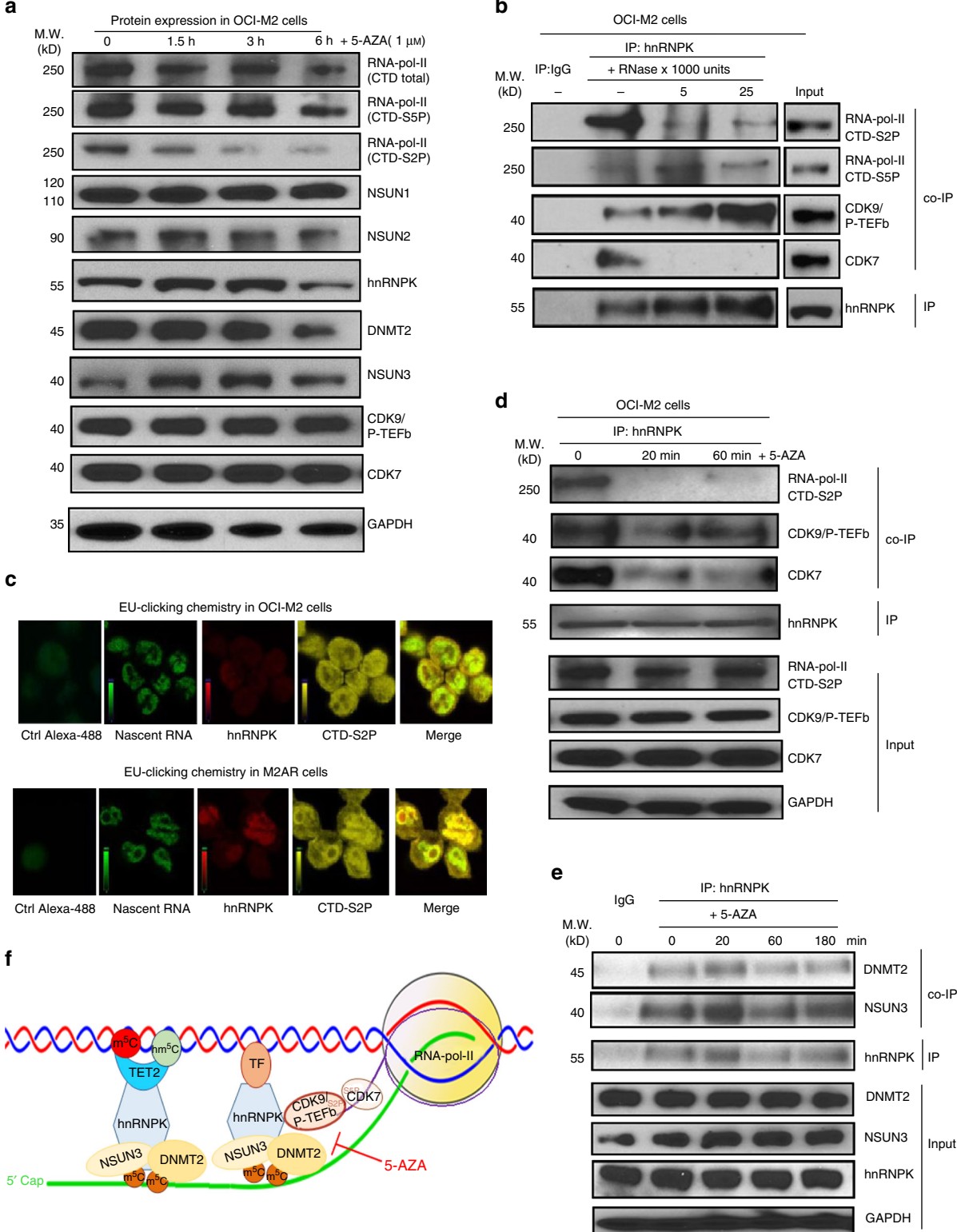

**Fig. 5** 5-AZA disrupts the interaction between active RNA-pol-II complex and hnRNPK/RCMT complex. **a** Examination of the effects of various epigenetic drugs on the growth of OCI-M2 and SC leukaemia cells. **b** Examination of the interactions between hnRNPK and the components of active RNA-pol-II complex and their RNA-dependence by RNase digestion-coupled co-IPs in OCI-M2 cells. **c** Elucidation of the subcellular localization and co-localization of hnRNPK with active RNA-pol-II CTD-S2P at nascent RNA in OCI-M2 cells by 5-ethynyluridine (EU) clicking chemistry STED confocal microscopy in OCI-M2 and M2AR cells. **d** Examination of the effects of 5-AZA on the interactions between hnRNPK and the components of active RNA-pol-II complex, including RNA-pol-II CTD-S2P, RNA-pol-II S5P, CDK7 and CDK9/P-TEFb in OCI-M2 cells treated with 1 μM 5-AZA for various times. **e** Examination of the effects of 5-AZA on the interactions between hnRNPK and DNMT2 as well as NSUN3 by co-IPs in OCI-M2 cells treated with 1 μM 5-AZA for various times. **f** A schematic presentation of the hnRNPK/RCMT-associated, 5-AZA-sensitive, active chromatin structure at nascent RNA in 5-AZA-sensitive leukaemia cells

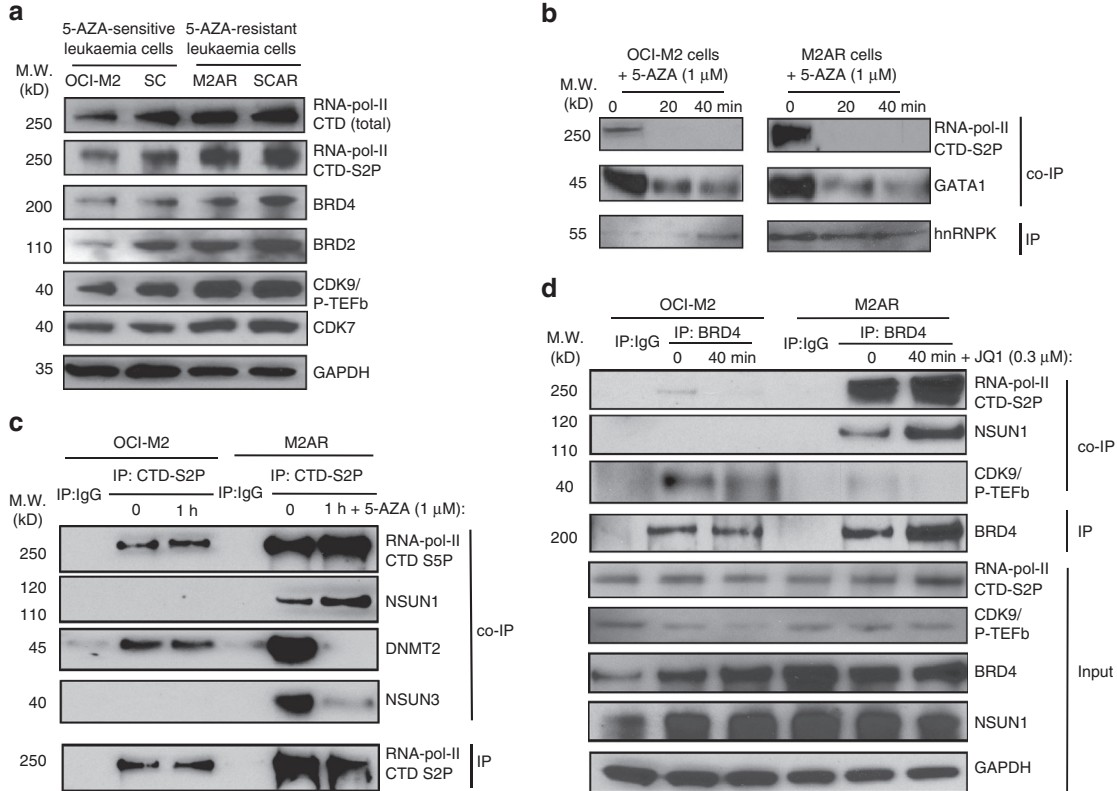

**Fig. 6** Identification of unique interactions between RNA-pol-II CTD-S2P and NSUN1 as well as BRD4 in 5-AZA-resistant leukaemia cells. **a** Examination of the protein expression of the components of active RNA-pol-II complex, BRD4 and BRD2 in the 5-AZA-sensitive OCI-M2 and SC cells and the 5-AZA-resistant M2AR and SCAR cells by western blotting. **b** Examination of the effects of 5-AZA on the interactions between hnRNPK and RNA-pol-II CTDS2P as well as GATA1 in OCI-M2 and M2AR cells by IP and co-IP. **c** Elucidation of a unique 5-AZA-resistant interaction between RNA-pol-II CTD-S2P and NSUN1 in the 5-AZA-resistant M2AR cells by IP and co-IP. **d** Elucidation of unique interactions between BRD4 and RNA-pol-II CTD-S2P as well as NSUN1 in the 5-AZA-resistant M2AR cells by IP and co-IP and the effects of JQ1 on these interactions

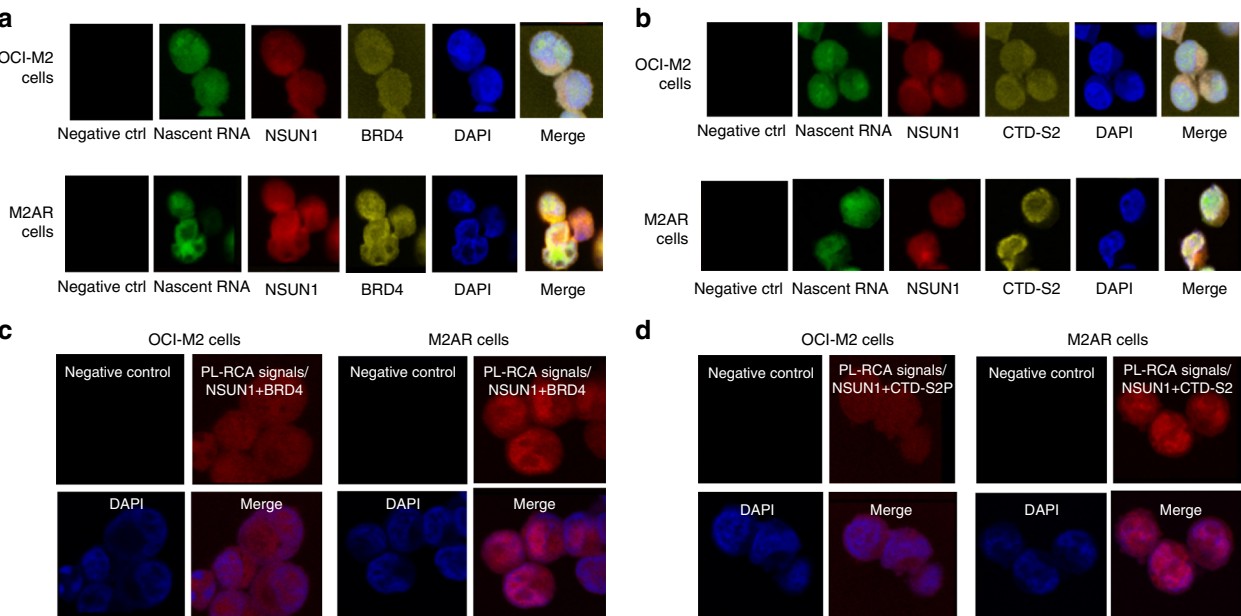

**Fig. 7** Distinct NSUN1/BRD4-associated active chromatin structures in 5-AZA-restsant leukaemia cells. **a** Elucidation of the subcellular localization of nascent RNA, NSUN1 and BRD4 by 5-ethynyluridine (EU)-clicking chemistry-coupled STED confocal microscopy in OCI-M2 and M2AR cells with 0.3 mM EU for 5 h. **b** Elucidation of the subcellular localization of nascent RNA, NSUN1 and RNA-pol-II CTD-S2P by EU-clicking chemistry-coupled STED confocal microscopy in OCI-M2 and M2AR leukaemia cells treated with 0.3 mM EU for 5 h. **c** Demonstration of markedly increased co-localization of NSUN1 and BRD4 by proximity ligation and rolling circle amplification (PL-RCA) in OCI-M2 and M2AR cells. **d** Demonstration of markedly increased co-localization of NSUN1 and RNA-pol-II CTD-S2P by proximity ligation and rolling circle amplification (PL-RCA) in OCI-M2 and M2AR cells

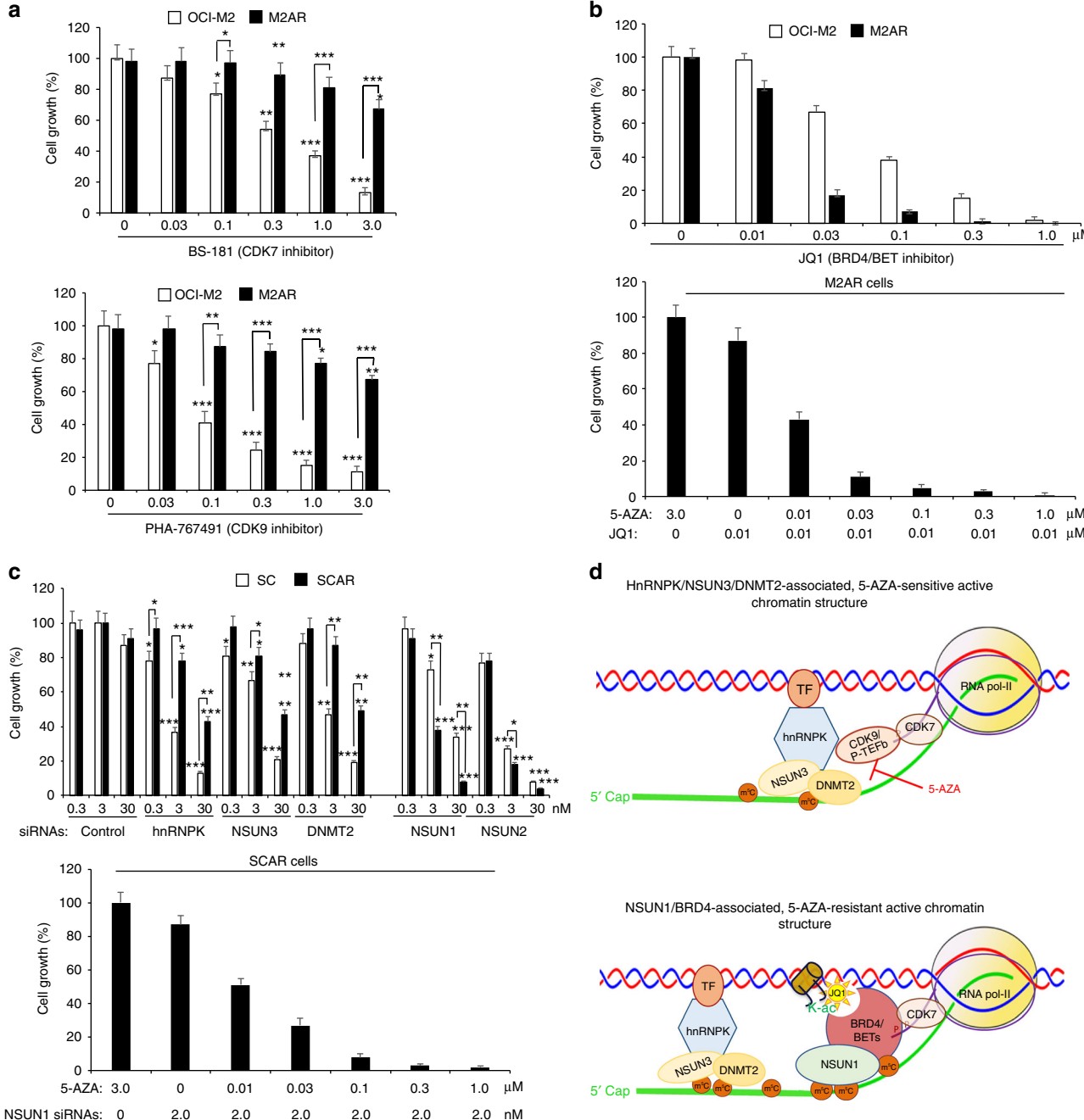

**Fig. 8** Differential growth inhibition with CDK7/9 inhibitors, BRD4 inhibitor JQ1 and RCMT siRNAs in 5-AZA-sensitive vs. 5-AZA-resistant leukaemia cells. **a** Measurements of the cell growth rates in OCI-M2 and M2AR cells treated with the CDK7 inhibitor, BS-181, and the CDK9 inhibitor, PHA-767491, at various concentrations for three days. **b** Measurements of the growth rates of OCI-M2 and M2AR cells treated with JQ1 at various concentrations for 3 days and the synergic growth inhibition in the M2AR cells treated with a low dose (0.01 μM) of JQ1 plus 5-AZA at various concentrations. **c** Measurements of the growth inhibition by the siRNAs targeting various RCMTs in SC and SCAR for 3 days and the synergic growth inhibition in the SCAR cells treated with 2 nM NSUN1 siRNAs plus 5-AZA at various concentrations. **d** A schematic presentation of the proposed working model of distinct 5-AZA responsive/resistant chromatin structures in 5-AZA-sensitive and 5-AZA-resistant leukaemia cells

data suggested that mRNA:m5C may play an important role in mediating 5-AZA resistance in MDS and AML.

**NSUN1/BRD4-associated active chromatin in clinical specimens.** We performed the PL-RCA on the 5-AZA-sensitive and 5-AZA-resistant AML/MDS bone marrow specimens. This experiment demonstrated very strong PR-RCA signals of co-localization of NSUN1 and RNA-pol-II CTD-S2P in the 5-AZA-resistant AML/MDS bone marrow cells and much weaker

PL-RCA signals in the 5-AZA-sensitive AML/MDS bone marrow cells (Fig. 10a). Quantitative analysis of the PL-RCA signal strength by manually counting the percentages of those PL-RCA-positive cells per 500 DAPI-positive cells demonstrated a significant increase in the PL-RCA signals of co-localization of NSUN1 and RNA-pol-II CRD-S2P in the 5-AZA-resistant AML/MDS bone marrow specimens compared to the 5-AZA-sensitive AML/MDS bone marrow specimens. The PL-RCA experiments with anti-NSUN1 and BRD4 antibodies performed in the same

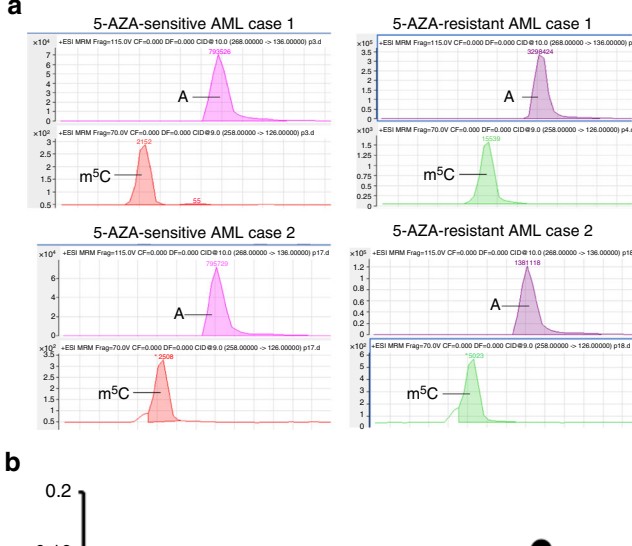

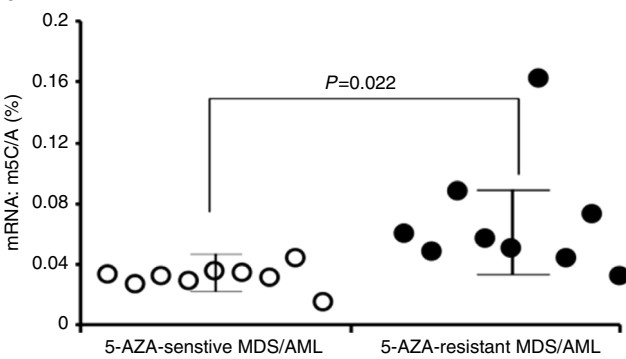

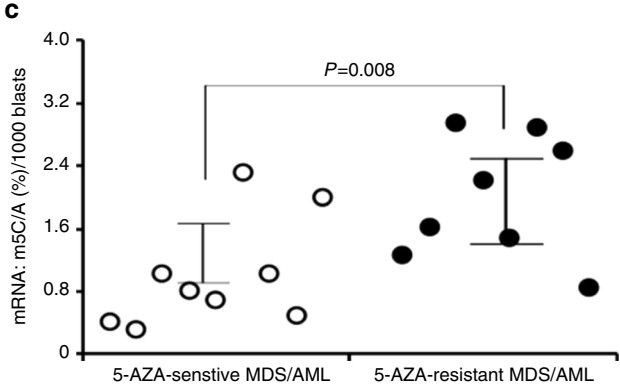

**Fig. 9** Mass spectrometric analysis of mRNA:m⁵C in clinical 5-AZA-sensitive and 5-AZA-resistant AML/MDS specimens. **a** Representative mass spectra of the mRMA:m⁵C and adenosine peaks using the Ficoll-Paque™ PLUS-purified mononuclear bone marrow cells from 5-AZA-sensitive and 5-resistant AML and MDS specimens. **b** Quantitative measurement of the m⁵C/A ratios in 5-AZA-sensitive and 5-AZA-resistant clinical AML/MDS bone marrow cells. **c** Comparison of the manual blast count-normalized mRNA:m⁵C levels in 5-AZA-sensitive vs. 5-AZA-resistant clinical AML/MDS bone marrow cells

sets of clinical bone marrow specimens demonstrated a similar co-localization, with significantly increased PL-RCA signals in the 5-AZA-resistant MDS/AML specimens compared to those in the 5-AZA-sensitive AML/MDS bone marrow specimens (Fig. 10b). We could not perform co-IP and EU-clicking chemistry studies because of insufficient numbers of intact leukaemia cells in the archived bone marrow specimens. We performed morphology, immunohistochemistry and STED confocal microscopy to investigate the expression and the subcellular localization of hnRNPK, NSUN1 and BRD4 in the pre-treatment clinical bone

marrow specimens from normal controls and from the patients with cytogenetically normal MDS, specifically, refractory anaemia with multilineage dysplasia (RCMD) subtype, and AML. Ten cases were selected for each group. We assessed the morphology and manual differential count of the bone marrow cells because blasts in erythroid and monocytic AML cases are often negative for CD34, a useful marker for identifying haematopoietic stem cells and leukaemic blasts. Immunohistochemical analysis demonstrated lower expression levels of CD34, hnRNPK, NSUN1 and BRD4 in the normal bone marrow cells, and significantly increased expression levels of these proteins in the bone marrow cells of the MDS/RCMD and AML cases, and the increased expression of these proteins was correlated with the disease progression in these MDS/AML cases (Fig. 10c). STED confocal microscopic analysis of clinical bone marrow specimens demonstrated increased expression and co-localization of hnRNPK and RNA-pol-II CTD-S2P in the AML bone marrow cells compared to normal bone marrow cells (Fig. 10d). Taken together, the PL-RCA and immunohistochemical data demonstrated significantly increased co-localization of NSUN1, BRD4 and RNA-pol-II CTD-S2P in 5-AZA-resistant MDS/AML cases and a disease progression-associated increase in expression of hnRNPK, NSUN1 and BRD4 in clinical MDS/AML specimens, supporting the clinical importance and relevance of RCMTs and hnRNPK as well as their associated chromatin structures in regulating 5-AZA resistance and disease progression in clinical MDS/AML.

**NGS to identify mutations in ASLC and ARLC lines**. Next generation sequencing (NGS) analysis of the 5-AZA-sensitive OCI-M2 and the 5-AZA-resistant M2AR cells identified 13,043 gene mutations/alterations (Supplementary Fig. 11a), which were not present in the 1000 Genomes Project database[43]. Gene ontology (GO) analysis was performed using the GO terms[44], including GO:0040029 (epigenetic regulators), GO:0006304 (DNA modifiers) and GO:0009451 (RNA modifiers), to identify the gene mutations involving epigenetic regulators, DNA and RNA modifiers, respectively. Among these mutated genes, 16 RNA modifier-encoding genes were unique to the 5-AZA-resistant M2AR leukaemia cell line (Supplementary Fig. 12a,b). Similar analysis of SC and SCAR cell lines identified 12,523 gene mutations/alterations, of which 15 involved RNA modifier-encoding genes that were unique to the SCAR cell line (Supplementary Fig. 12c,d). However, we have not been able to verify the clinical significance of these gene mutations and their impact on the 5-AZA response/resistance-associated active chromatin structures at this point.

## Discussion
DNA hypomethylating agents have been widely used to treat various haematologic and non-haematologic malignancies. However, accumulating evidence has demonstrated that DNA methylation status does not correlate with clinical response to hypomethylating agents[45]. In fact, MDS/AML patients carrying inactivating *DNMT3A* mutations respond better to the DNMT inhibitors, such as 5-AZA and decitabine, than those patients having wild-type *DNMT3A*[46, 47]. A majority (about 90%) of 5-AZA is incorporated into RNA, and only a small portion (about 10%) of 5-AZA is incorporated into DNA[32]. These data suggest that 5-AZA may work through an RNA-mediated mechanism. Currently, little is known about the physiological functions and the pathological roles of RNA:m⁵C and its writers, readers and erasers. Our data demonstrate that hnRNPK directly interacts with a subset of RCMTs, namely DNMT2 and NSUN3, to form a functional complex important for the integrity of these proteins

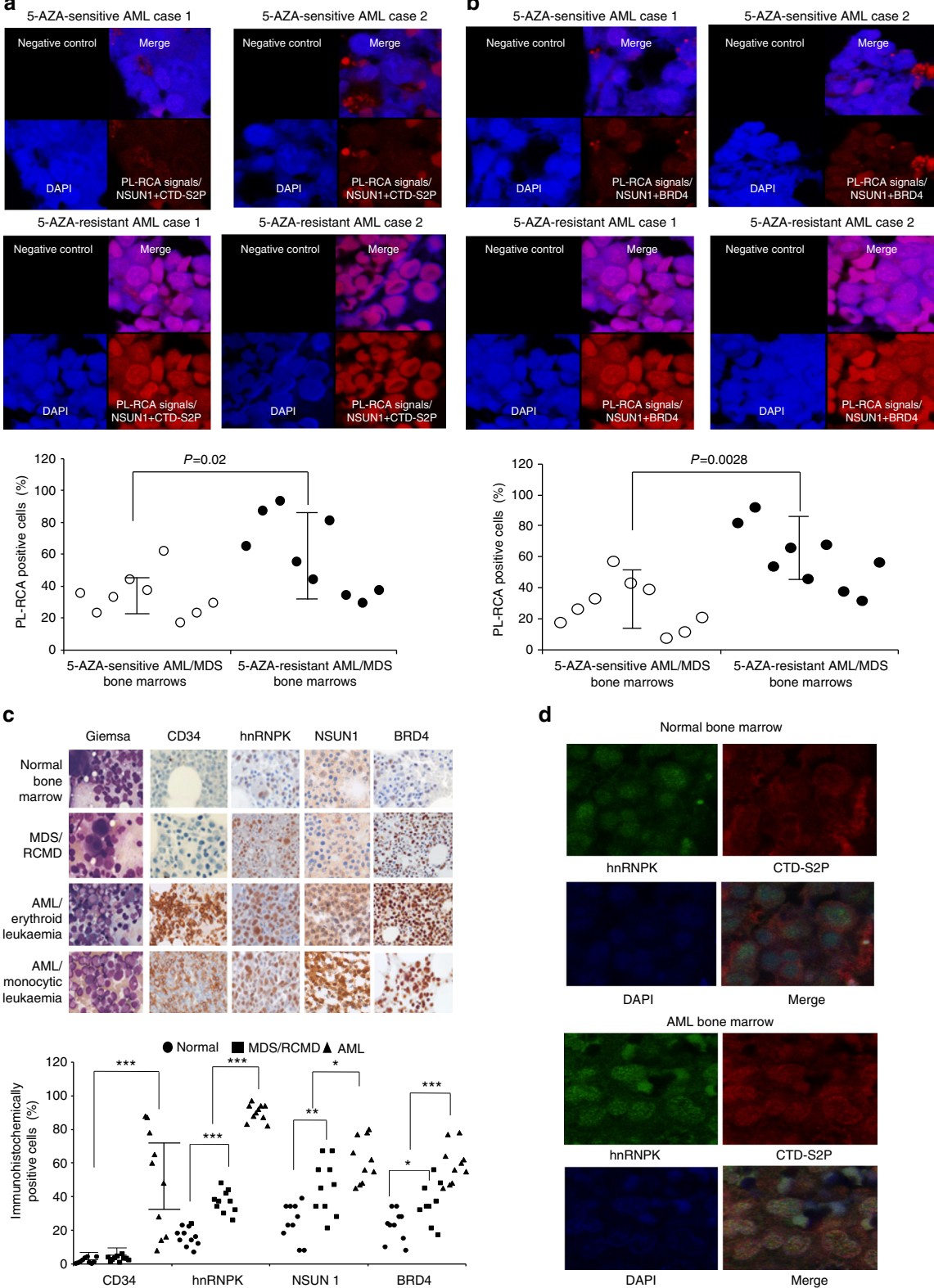

**Fig. 10** Increased expression of hnRNPK, NSUN1 and BRD4 as well as their co-localization with active RNA-pol-II in 5-AZA-resistant AML/MDS specimens. **a** Visualization and quantitative measurement of co-localization of NSUN1 and RNA-pol-II CTD-S2P in 5-AZA-sensitive and 5-AZA-resistant clinical AML/MDS bone marrow specimens by PL-RCA. **b** Visualization and quantitative measurement of co-localization of NSUN1 and BRD4 in 5-AZA-sensitive and 5-AZA-resistant clinical AML/MDS bone marrow specimens by PL-RCA. **c** Morphological and immunohistochemical studies of the expression of CD34, hnRNPK, NSUN1 and BRD4 in the bone marrow cells from normal controls, cytogenetically normal MDS/refractory cytopenia with multilineage dysplasia (RCMD), and AML, $n = 10$ for each group, with an Olympus BX43 microscope at ×400 magnification. **d** STED confocal microscopic analysis of the expression and subcellular localization of hnRNPK and RNA-pol-II CTD-S2P in normal and AML bone marrows

and survival of leukaemia cells (Fig. 1). hnRNPK also selectively interacts with the lineage-determining factors, GATA1 and SPI1/PU.1, and the DNA demethylase TET2 as well as CDK9/P-TEFb that recruits RNA-pol-II to nascent RNA to form 5-AZA-sensitive active chromatin structure (Figs. 2–5). This study is, to the best our knowledge, the first study that demonstrates RCMTs/hnRNPK-mediated lineage-associated drug-responsive chromatin structure in leukaemia cells.

Recent genome-wide studies have demonstrated widespread TF-trapping by RNA and extensive RNA-mediated interactions in gene regulatory elements[48–51]. However, the mechanistic details of TF-trapping by RNA remain unclear. Our data demonstrate that there are selective 5-AZA-/RNase-sensitive interactions between the lineage-determining factors and chromatin modifiers in the leukaemia cells (Fig. 4). To date, about 1500 RBPs with various affinities to different classes of RNA have been identified[52]. We are particularly interested in hnRNPs, a group of RBPs that are associated with nascent precursor mRNA (pre-mRNA) transcripts and play a critical role in regulating transcription and mRNA fate[48, 53–56]. We used antibodies against GATA1 (in OCI-M2) and SPI1/PU.1 (in SC) to screen a small pool of commercially available antibodies against hnRNPs in various hnRNP complexes[57, 58], including A0, A1, A2/B1, C1/C2, AUF1 and K. We identified hnRNPK as a unique RBP that directly interact with both GATA1 and TET2 in OCI-M2 cells (Fig. 4) as well as SPI1/PU.1 and TET2 in SC cells (Supplementary Fig. 6). Our data together suggest that RNA and hnRNPK mediate 5-AZA-sensitive interactions between the lineage-determining factors (GATA1 and SPI1/PU.1) and chromatin modifiers in the OCI-M2 and SC leukaemia cells.

One of the important findings in this study is that our data demonstrate the presence of 5-AZA-resistance-associated interactions between RNA-pol-II (CTD S2P) and NSUN1 as well as BRD4 and the formation of a distinct active chromatin structure at nascent RNA in ARLCs (Figs. 6–8). Previously published studies have demonstrated that BRD4 can interact with CDK9/P-TEFb through its bromodomain, resulting in CDK9/P-TEFb-dependent phosphorylation of the serine 2 residue in the CTD of RNA-pol-II and activation of transcription in vivo[59]. However, our data show that the increased BRD4- and NSUN1-associated RNA-pol-II CTD-S2P in the ARLCs is not associated with CDK9/P-TEFb (Fig. 6d), which is consistent with recent studies demonstrating BRD4 functions as an atypical kinase to directly bind and phosphorylate the serine 2 of RNA-pol-II CTD in vitro and in vivo under conditions where other CTD kinases are inactive[60]. This notion is further supported by the demonstration that ARLCs have a preferential growth inhibition towards the BRD4 inhibitor, JQ1, and to the downregulation of NSUN1 by siRNAs, whereas the original ASLCs are preferentially inhibited by the CDK9/CDK7 inhibitors and to the downregulation of HNRNPK, NSUN3 and DNMT2 siRNAs (Fig. 8 and Supplementary Fig. 11). Our data suggest a model of distinct RNA:m5C and RCMT-mediated active chromatin structures in 5-AZA-sensitive vs. ARLCs (Fig. 8e). Specifically, in ASLCs, hnRNPK directly interacts with NSUN3, DNMT2 and CDK9/P-TEFb to recruit active RNA-pol-II, leading to the formation of 5-AZA-sensitive, active chromatin structure. In contrast, in ARLCs, NSUN1, but not hnRNPK, interacts with BRD4 and RNA-pol-II CTD-S2P to form an active chromatin structure that is resistant to 5-AZA, but highly sensitive to the inhibition of BRD4 and NSUN1 (Figs. 6–7 and Supplementary Figs. 7–9). The immuno-histochemistry, mass spectra and PL-RCA on the MDS/AML bone marrow specimens confirm the clinical importance and relevance of RNA:m5C and RCMT-associated active chromatin structures in 5-AZA resistance in clinical MDS/AML (Figs. 9–10). We performed NGS to try to identify mutations/alterations that

are responsible for formation of the NSUN1-/BRD4-associated 5-AZA-resistant chromatin structure in leukaemia cells (Supplementary Fig. 12). However, at this point we have not been able to identify such gene mutations. Additional studies are needed to identify the genetic alterations and the chemical modifications underlying the RCMTs-associated drug responsive/resistant chromatin structures in leukaemia cells and to determine the potential and the limitation of using RNA:m5C/RCMTs and their associated chromatin structures as diagnostic biomarkers and therapeutic targets in MDS/AML and other haematologic and non-haematologic malignancies/diseases.

## Methods

**Clinical specimens and IRB.** The immunohistochemical and confocal microscopic studies were performed using the archived bone marrow specimens and approved by the Institutional Review Boards (IRB-14-0512). The detailed information about the diagnoses, immunophenotypes, cytogenetic and gene mutation profiles of the clinical cases are provided in Supplementary Table 2.

**Cell culture and reagents.** Ten cell lines, namely OCI-M2 (ACC-619, DSMZ), K562 (CCL243, ATCC), MonoMac6 (ACC-124, DSMZ), SC (CRL-9855, ATCC), THP1 (TIB202, ATCC), U937(CRL-1593.2, ATCC), MV4-11 (CRL9591, ATCC), HL-60 (HL-60, ATCC), KASUMI-1 (CRL-2724, ATCC) and NB4 (CCL-246, ATCC), were used in the initial drug screening experiments. 5-AZA-resistant cells, M2AR and SCAR, were developed from 5-AZA-sensitive OCI-M2 and SC, respectively, in the lab, and OCI-M2, SC, M2AR and SCAR were used primarily in this study using standard culture conditions. The detailed information about the cell lines is provided in Supplementary Table 1 and the experimental reagents used in this study and their resources are provided in Supplementary Table 3.

**Isolation of mononuclear cells from MDS and AML bone marrows.** The mononuclear cells were purified from the 5-AZA-sensitive and 5-resistant AML and MDS bone marrow aspirate specimens using the GE Ficoll-Paque™ PLUS (Cat. No. Sigma-GE 17-1440, Sigma-Aldrich company) by following the manufacturer's protocol. Total RNA was extracted from the mononuclear cells with Invitrogen™ TRIzol™ Reagent, and mRNA was further purified with two rounds of the NEB polyA Spin™ purification. For m5C quantitation, approximately 50 ng of RNA was digested with 0.5 unit of nuclease P1 from Penicillium citrinum (Sigma, N8630) in 13 μl 25 mM NaCl and 2.5 mM $ZnCl_2$ buffer for 1 h at 42 °C. One unit FastAP and 10× FastAP buffer were then added to the reaction for another 3 h incubation at 37 °C. After digestion, the reactions were diluted to 50 μl and subjected to liquid chromatography tandem-mass spectrometry (LC-MS/MS, Agilent 6460).

**Preparation of nuclear extracts from leukaemia cells.** Nuclear extracts were prepared by using a protocol modified from previously published methods[61, 62]. 1–2×10^6 mononuclear leukaemia cells were centrifuged at 500 × $g$ for 5 min. The cells were then washed with PBS, and resuspended in the cell lysis buffer (10 mM HEPES; pH 7.5, 10 mM KCl, 0.1 mM EDTA, 1 mM dithiothreitol (DTT), 0.5% Nonidet-40 and 0.5 mM PMSF along with the protease inhibitor cocktail (Sigma)) and allowed to swell on ice for 20 min with intermittent mixing. Tubes were vortexed to disrupt cell membranes and then centrifuged at 12,000 × $g$ at 4 °C for 10 min. The pelleted nuclei were washed once with the cell lysis buffer and resuspended in the nuclear extraction buffer containing 20 mM HEPES (pH 7.5), 400 mM NaCl, 1 mM EDTA, 1 mM DTT, 1 mM PMSF with protease inhibitor cocktail and incubated in ice for 30 min. Nuclear extract was collected by centrifugation at 12,000 × $g$ for 15 min at 4 °C.

**Western blot and immunoassays.** Western blotting, IP, co-IP and dot blotting for analysis of protein expression, protein−protein interactions and RNA/DNA 5-mC were performed by following the standard protocols[63]. Proximity ligation and rolling circle amplification (PL-RCA)-coupled confocal microscopy were performed by using Duolink® in situ detection reagents (Cat. No. DUO92008, Sigma-Aldrich) for in vivo analysis of expression and co-localization of hnRNPK and other proteins. The catalogue numbers of the vendors of the antibodies used in the PL-RCA experiments are listed in the supplementary information. The in situ hybridization and PL-RCA were performed by following the manufacturer's protocols.

**Modified chromatin immunoprecipitation assay.** Cultured leukaemia cells and bone marrow mononuclear cells from clinical specimens were used. Five million mononuclear cells were used to perform a single regular ChIP. For sequential ChIP, 10 million mononuclear cells were used. The ChIP protocol for quantitative analysis of locus-specific 5-mC and 5-hmC as well as RNA-pol-II recruitment was adopted and modified from the original protocol[64]. We modified the original protocol developed by introducing an acid-based elution buffer, which enabled

direct quantitative PCRs (Q-PCRs) to be performed on eluates without additional purification. The signals at specific gene locus (SPI1/PU.1) were analysed by qPCR. The detailed ChIP protocol is described in the Supplementary Methods. The sequences of the qPCR primers are provided in Supplementary Table 4. The following is the detailed modified ChIP protocol.

Step 1. Preparation of bone marrow mononuclear cells for ChIP

1. One millilitre (ml) was mixed with 5 ml of phosphate buffered saline (PBS) in a 15-ml conical tube.
2. The mixed bone marrow/blood was delicately overlaid onto 5.0 ml of Ficoll gradient Histopaque-1077 (Sigma #1077-1, St. Louis, MO).
3. Cellular components were separated by centrifuge at $400 \times g$ for 20 min at room temperature, and the mononuclear layer at the interface between the two solutions was collected.
4. Ten millilitres of PBS were added to the mononuclear cells and then centrifuged at $500 \times g$ for 5 min.
5. Cell pellets were resuspended in PBS, and the cells were quantified and adjusted to a final concentration of 5 million cells per ml in PBS.

Step 2. Cross-linking DNA/histone complexes

1. Add 10% formaldehyde/PBS (pH = 7.5) directly to tissue culture media to a final concentration of 1% formaldehyde.
2. Stop the cross-linking reaction by adding 1.25 M glycine to a final concentration of 0.125 M glycine. Gently rock the cells at room temperature for 5 min.
3. Collect the cell suspension and scrape adherent cells into a 50-ml conical tube. Centrifuge the cells at $800 \times g$, 4 °C for 5 min.
4. Discard the supernatant and wash the pellet once with cold PBS plus 1× protease inhibitor cocktail (from Roche). Centrifuge the cells at 2500 rpm, 4 °C for 5 min.
5. Discard the supernatant and re-suspend the cells with cold PBS plus 1× protease inhibitor cocktail (from Roche), adjusting the volume to $10 \times 10^6$ cells per ml.
6. Aliquot the cells in Eppendorf tubes ($10 \times 10^6$ cells/ml/tube). Centrifuge the cells at $800 \times g$, 4 °C for 5 min.
7. Discard the supernatant. Snap-freeze the cell pellets in dry ice and store the cell pellets in −80 °C.

Step 3. Chromatin immunoprecipitation

1. Thaw two tubes of cell pellets on ice and re-suspend the cell pellets in cell lysis buffer ($10 \times 10^6$ cells/200 µl) plus 1× protease inhibitor cocktail.
2. Centrifuge the cells at $2500 \times g$ for 5 min at 4 °C to pellet the nuclei.
3. Discard the supernatant and resuspend the nuclei vigorously in nuclei lysis buffer plus 1× protease inhibitor cocktail ($10 \times 10^6$ cells/100 µl nuclei lysis + 1× protease inhibitor). Incubate the lysates on ice for 10 min.
4. Transfer the nuclear lysates to a 4 ml sonication tube and add 1 ml chromatin immunoprecipitation buffer with 0.4% of Triton X-100 (IP buffer) plus 1× protease inhibitor cocktail. Resuspend the nuclear lysates vigorously using a pipette.
5. Sonicate the chromatin to an average length of 300–500 bp while keeping the samples on ice (the time and number of pulses will vary depending on the sonicator, cell type and extent of crosslinking). The sonication condition was 4× cycles, 30 s/cycle, 40% output and 2 s/pulse.
6. Carefully remove the supernatant and transfer to a new tube. To reduce non-specific background, pre-clean chromatin by adding 150 µl of protein A Sepharose beads. Incubate the sample on a rotating platform at 4 °C for 1.5 h. Next, spin down and remove the protein A beads from the sample.
7. Save the supernatant, add 2 ml of IP buffer and mix well. The total volume should be 3.2 ml. Save 200 µl as the input control.
8. Equally divide the samples into specific antibodies and an equal number of IgG controls. The antibody concentrations needed for ChIP vary greatly.
9. Incubate the sample with constant rotation overnight at 4 °C.
10. Add 70 µl of salmon sperm DNA/protein A/G agarose or DNA/protein A/G agarose (50% slurry cells, EMD Millipore).
11. Incubate the sample at constant rotation for 2 h at 4 °C.
12. Pellet the agarose by gentle centrifugation ($200 \times g$ at 4 °C, 4 min). Carefully remove the supernatant containing unbound, non-specific DNA.
13. Wash the protein A agarose/antibody/complex for 3 min on a rotating platform with 1 ml of each of the buffers containing 1× protease inhibitor cocktail (freshly added), listed in the order given below:
    Low Salt Immune Complex Wash Buffer, one wash
    High Salt Immune Complex Wash Buffer, one wash
    LiCl Immune Complex Wash Buffer, one wash
    1× TE, two washes.
14. Save the pellets and proceed to elution.

Step 4. Elution of immunoprecipitated chromatin.

1. Add 200 µl of the low pH elution buffer to the beads and gently re-suspend by pipetting several times in ice without vortexing.
2. Microcentrifuge at $200 \times g$ for 2 min and carefully transfer the supernatant fraction (eluate) to another tube on ice.

3. Immediately add 20 µl of the high pH buffer to the 200 µl eluate to neutralize the pH of the eluate (on ice).
4. Each ChIP should have a total volume of 220 µl.
5. Use 2–4 µl of the eluates for Q-PCRs.

Step 5. Direct Q-PCR on the neutralized elutes:

1. Save 220 µl of the 440 µl total elute for Q-PCR.
2. Use 2−4 µl of elute per Q-PCR reaction (25 µl). Set the Q-PCR conditions according to the optimal annealing temp and DNA concentration vs. background noise. We found that 1–2 µl of input should give a strong signal.

Cell Lysis buffer: 5 mM PIPES pH 8.0, 85 mM KCL, 0.5% NP40, 1 × protease inhibitor cocktail (add prior to use)

Nuclei Lysis buffer: 50 mM Tris-HCl pH 8.1, 10 mM EDTA, 1% SDS, 0.5% Empigen BB (30%) (optional), 1× protease inhibitor cocktail (add prior to use)

IP Dilution buffer: 0.5% Triton X-100, 1.2 mM EDTA, 16.7 mM Tris-HCl pH 8.1, 167 mM NaCl, 1× protease inhibitor cocktail (add prior to use)

Low Salt Wash Buffer: 0.5% Triton X-100, 2 mM EDTA, 20 mM Tris-HCl pH 8.1, 150 mM NaCl.

High Salt Wash Buffer: 0.5% Triton X-100, 2 mM EDTA, 20 mM Tris-HCl pH 8.1, 500 mM NaCl.

LiCl Wash Buffer: 0.25 M LiCl, 0.5% IGEPAL CA630, 1 mM EDTA, 10 mM Tris pH 8.1

TE buffer: 10 mM Tris-HCl pH 7.5, 1 mM EDTA (Make from 1 M stock of Tris-HCl pH 7.5) and 500 mM stock of EDTA (pH 8.0).

Low pH elution buffer: 0.125 M glycine pH 2.5 (pH value must be accurately measured!)

High pH neutralizing buffer: 1 M Tris pH 9.0 (pH value must be accurately measured!)

Preparation of Salmon Sperm DNA protein A −50% slurry cells: 0.75 ml protein A beads, 600 µg sonicated salmon sperm DNA, 1.5 mg BSA.

**Crosslink-assistant DNA modification immunoprecipitation assay.** Crosslink-assisted DNA modification IP assay was modified from the previously published methylated DNA IP protocol[65]. DNA was isolated from cultured cells or mononuclear bone marrow cells using a Qiagen DNAeasy kit (catalogue # 69506, Valencia, CA). Two micrograms of isolated DNA were used for each experiment. The DNA (2 µg) was incubated with 3 µl of *Nla*III restriction enzyme (10,000 units/ml) 37 °C overnight in the CutSmart™ Buffer (NEB Inc., Ipswich, MA), and the enzyme was inactivated by incubation at 65 °C for 20 min. The fragmented DNA was then incubated with anti-5-mC and anti-5-hmC antibodies in IP buffer for 6 h at 4 °C. The antibody/DNA complexes were crosslinked by 0.1% formaldehyde at 4 °C for 10 min, and 125 mM glycine was used to terminate the crosslinking. The crosslinked antibody/5-mC or 5-hmC complexes were further diluted and incubated with rabbit anti-mouse IgG (catalogue #06-371, Millipore Corporation, Billerica, MA). A modified protocol with acid buffer as described in the Supplementary Methods was used to isolate DNA for direct Q-PCRs and ligation-mediated PCRs (to assess local and global 5-mC and 5-hmC status). The sequences of the qPCR primers are provided in Supplementary Table 4.

**Chromosome conformation capture (3C).** The chromatin conformation capture (3C) qPCR protocol published by Dr. Dekker's lab[66] was followed. Briefly, 10 million mononuclear cells were used for each experiment and were cross-linked using 1% formaldehyde. Next, the *Nla*III restriction enzyme was used to digest the DNA into small fragments, which were then ligated using T4 DNA ligase overnight at 16 °C. A high temperature (96 °C for 10 min) was used to reverse the cross-links. Polymerase chain reaction (PCR) used primers against the site of ligation to semi-quantitatively assess the frequencies of a restriction fragment of interest, and quantitative PCR was performed using TaqMan probes (3C-qPCR) to more quantitatively measure the DNA fragments. The sequences of the qPCR primers are proved in Supplementary Table 4.

**Visualize and quantify the binding of hnRNPK to RNA.** The 6-FAM (Fluorescein)-labelled methylated and unmethylated RNA oligos were purchased from Integrated DNA Technologies (Skokie, IL 60076). The names and sequences of the RNA oligos are: CR1: FAM-rCrCrCrCrUrCrCrCrCrGrCrCrCrArCrGrCrCrCrU; mCR1: FAM-mCmCmCmCrUmCmCmCmCrGmCmCmCrAmCrGmCmCmCrU. Purified hnRNPK protein was purchased from LifeSpan Biosciences, Inc. (Seattle, WA).

For binding of purified hnRNPK to RNA oligos, various amounts of purified recombinant hnRNPK or nuclear lysate were incubated with methylated and unmethylated RNA oligos at 10 µM concentration and 4 µl of anti-hnRNPK antibody in 100 µl of binding buffer containing 20 mM HEPES (pH 7.5), 1 mM DTT, 1.2 mM EDTA, 16.7 mM Tris-HCl pH 8.1, 167 mM NaCl, 1× protease inhibitor cocktail at 4 °C for 30 min, with gentle rotation. Then 30 µl of protein A agarose beads were added and incubated at 4 °C for an additional 30 min with gentle rotation. The beads were washed with low salt wash buffer containing 0.5% Triton X-100, 2 mM EDTA, 20 mM Tris-HCl pH 8.1, 150 mM NaCl and high salt wash buffer containing 0.5% Triton X-100, 2 mM EDTA, 20 mM Tris-HCl pH 8.1, 500 mM NaCl, for 5 min twice. The FAM-RNA oligo/hnRNPK/antibody-agarose

beads were resuspended in 10 μl of the binding buffer and the beads were dropped onto glass slides for STED confocal imaging microscopy examination. A Leica SP5 II Stimulated Emission Depletion (STED) Continuous Wave (CW) Super-resolution Laser Scanning Confocal microscopy at the Integrated Microscopy Core Facility of University of Chicago was used for confocal imaging analysis.

**In vivo transfection assay.** Lipofectamine™ RNAiMAX Reagent (Cat. 13778075, Invitrogen) was used to transfect leukaemia cells with siRNAs targeting hnRNPK and various RCMTs. The transfection assays were performed by following the company-provided forward transfection protocol for bone marrow cells. The KDalert™ GAPDH Assay Kit (Cat. AM1639, Invitrogen) was used to optimize transfection conditions and to assess for transfection efficiency by following the company-provided protocol. The sequences of the siRNAs and their resources are provided in Supplementary Table 5.

**Proximity ligation and rolling cycle amplification.** For cultured leukaemic cell lines and clinical bone marrow cells, 5 million mononuclear cells were used for each in vivo proximity ligation rolling cycle amplification (PL-RCA) experiment. The cells were pre-treated with fixation and permeabilization, and then deposited on polylysine-coated glass slides for PL-RCA. Routinely processed sections (3 μM in thickness) of decalcified paraffin-embedded clinical bone marrow core biopsy specimens were also used for PL-RCA after deparaffinization and rehydration with xylene and ethanol and antigen retrieval with 0.05% trypsin treatment for 20 min. The Duolink® in situ kit (catalogue # DUO92101, Sigma, St. Louis, MO) was used to study co-localization of RNA-pol-II, RCMTs, hnRNPK and BRD4 by following the manufacturer's protocol for in vivo PL-RCA.

**In vivo analysis of nascent RNA by EU-clicking chemistry.** Five million leukaemic cells were treated with 0.3 mM ethylene uridine (EU) and harvested at various time points (1, 2, 3 and 6 h). Click-iT® Nascent RNA Capture Kit (catalogue # C10330, ThermoFisher Scientific, Walthan, MA) was used to visualize and quantify nascent RNA in vivo by following the manufacturer's protocol that is adopted from the previously published 5-ethynyluridine (EU)-labelling protocol[37].

**Gene knockdown using siRNAs and cell transfection assay.** The siRNAs for knockdown of HNRNPK, NSUN1, NSUN2, NSUN3 and DNMT2 were purchased from OriGene Technologies Inc. The sequences of the siRNAs are provided in the Supplementary Information. Lipofectamine® RNAiMAX™ Transfection Reagent from ThermoFisher Scientific company was used for optimization of delivery of the siRNAs in the 5-AZA-sensitive (SC and OCI-M2) and the 5-AZA-resistant (SCAR and M2AR) leukaemia cells. The cells were grown to 70% confluence, serum-starved for 3 h and then transfected with the siRNAs and Lipofectamine® RNAiMAX™ Transfection Reagent following the manufacturer's protocols.

**Mass spectrometric analysis of mRNA:m5C.** The mononuclear cells were purified from the 5-AZA-sensitive and 5-AZA-resistant AML and MDS bone marrow aspirate specimens with the GE Ficoll-Paque™ PLUS (Cat. No. Sigma-GE 17-1440, Sigma-Aldrich company) as described in the Supplementary Methods. Total RNA was extracted from the mononuclear cells with Invitrogen™ TRIzol™ Reagent, and mRNA was further purified with two rounds of the NEB polyA Spin™ purification. For m5C quantitation, approximately 50 ng of RNA was digested with 0.5 unit of nuclease P1 from Penicillium citrinum (Sigma, N8630) in 13 μl 25 mM NaCl and 2.5 mM $ZnCl_2$ buffer for 1 h at 42 °C. One unit FastAP and 10× FastAP buffer were then added to the reaction for another 3 h incubation at 37 °C. After digestion, the reactions were diluted to 50 μl and subjected to liquid chromatography tandem-mass spectrometry (LC-MS/MS, Agilent 6460).

**Data availability.** The data sets generated during the current study are available from the corresponding author on reasonable request.

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

## Acknowledgements

We thank Dr. Vytas Bindokas and the UChicago Integrated Microscopy Core Facility staff for their technical support, Dr. Pieter Faber and the UChicago Genomic Facility staff for the help with NGS, Dr. Shihong Li and Ms. Can Gong at the UChicago Human Tissue Resource Center for the help with immunohistochemistry, Dr. Yali Dou in Department of Pathology, University of Michigan at Ann Arbor for providing MM401, the faculty members of the Section of Haematology/Oncology and Section of Hemato-pathology, University of Chicago for their help. This study has been supported by the funding to J.X.C. from Cancer Research Foundation Young Investigator Award, an Institutional Research Grant (#IRG-16-222-56) from the American Cancer Society, the Cancer Center Support Grant (#P30 CA14599) of the University of Chicago Medicine Comprehensive Cancer Center, Swim Across America Rush University/University of Chicago, CTSA-ITA Core Subsidies from the University of Chicago ITM grant (#UL1TR002389) and research/education fund from Department of Pathology, University of Chicago.

## Author contributions

J.X.C. conceived this study, designed the experimental plan, performed some of the experiments, interpreted data and wrote the manuscript. L.C., Y.L. and M.Y. performed the IP, drug treatment and siRNA knockdown experiments. A.C. contributed to the clinical data. J.W. and C.H. contributed mass spectrometric analysis of mRNA:m5C/A, K. A.W. and Q.J.S. contributed sequencing data analysis, J.M.S. and J.A. contributed clinical specimens. R.A.L., M.M.L.B. and J.W.V. contributed to the project by advising on clinical, genetics, project direction and morphological analysis.

## Additional information

**Competing interests:** The authors declare no competing interests.

