## [Peer Review File · Nature Communications]

Reviewers' comments:

Reviewer #1 (Remarks to the Author):

The authors performed an interesting study to investigate the potential involvement of RNA/RNA modifications/RNA modifying and/or binding proteins in resistance towards 5-AZA in myeloid leukemia. Their data suggest that hnRNPK forms a complex with various RCTMs and that this complex is of importance for the survival of myeloid cell lines. Further, the authors suggest that BRD4 is involved in an atypical activation of RNA-pol-II (as previously has been shown for this protein) in 5-AZA resistant leukemias, and that JQ1 (or rather derivatives thereof that have a longer half-life in vivo; the referees comment) could be used in the treatment of 5-AZA resistant AML.

My concerns with this study are outlined below.

General points

1. It would be much easier to comment on the manuscript if the authors included page and/or line numbering.
2. The authors should have proofread the manuscript again prior to submission. There are now multiple instances of typos, duplication of words (e.g. "to measure to measure"), mistakes in various figures, etc., which give an unprofessional impression.

Other comments

1. Fig. 1a: Indicate which sample (left or right) that corresponds to OCI-M2 and SC, respectively, as well as M2AR1 and SCAR1.
2. Data related to Fig. 1b-c: Have the authors performed the corresponding assay for SC and SCAR1 as well? This should be done, and be included at least as a "Data not shown", depending on how interesting the results are.
3. Supp. Fig. 1b: Please add name of the cell line(s) in the legend and/or figure instead of only "...erythroid leukaemia cells...".
4. Supp. Fig. 2a: The authors have stated "Hypermethylation (5-hmC)" and "Methylation (5-mC)" in the top of this panel. Is that correct or should it refer to hydroxymethylcytosin and methylcytosin?
5. Supp. Fig. 2b, d and e: Y axis legend is missing.
6. Data related to Fig. 2:
 - Not stated here or in the corresponding Results section which cell line that was used for these experiments.
 - How many different siRNAs were evaluated per mRNA? At least two different ones are preferred to make sure any changes in cellular behavior etc. are not due to off-target effects. Especially for NSUN1/2 for which no knockdown was seen, additional siRNAs should be tested.
 - What is "100%" in Fig. 2c, since the control only reaches ~75%? These results suggest that even the control siRNA affects proliferation of the cells, which is a bit worrisome.
7. At the top of the fourth page in the Results section, the authors refer to Table 1. Should this be Supplementary Table 1?

8. Fig. 3: Why are the order of high versus low concentration the opposite in a) and b) (hnRNP amount) compared to c) and d) (lysate amount)? Consistency would be preferred.

9. Data related to Fig. 4: It is a bit bold to say that erythroid and monocytic cells have different chromatin structures at SPI1/PU1 etc. when only one cell line per cell type has been investigated. The differences could potentially be cell line specific and not cell type specific. Another erythroid and monocytic cell line should be investigated for the authors to be able to make that statement.

10. Supp. Fig. 3d: Both OCI-M2 and SC are mentioned in the figure legend, but only one bar per condition is shown in the figure. Is that an average of both cell lines or are only data from one of the cell lines shown? Data for both cell lines should be included in the figure in separate bars.

11. The heading of Fig. 6 and corresponding heading in the Results section: The authors only show (and mention) data generated using OCI-M2, and can thereby not include the "erythroid vs. monocytic leukaemia" in the headings. For this, at least one (or preferably more) monocytic cell line(s) should be investigated. Further, three of the lower panels in a) are identical to three of the left hand panels in b) – please avoid this kind of redundancy!

12. Fig. 7: The RNase amounts stated in all of the related panels suggest that duplicates of the different concentrations (5, 5, 25, 25) have been investigated on the gel/WB, but the corresponding results do not quite fit with what would be expected in that case - for instance are there highly different results for TET2 in the two different lanes stated to show data after treatment with 25K units of RNase in the top panel of a). Further, the blot for the TET2 co-IP in b) show all but trustable duplicates. Should it be a titration of four different concentrations and not only duplicates of two different concentrations? If this is a misunderstanding of the figure, that figure and corresponding legend need to be clarified quite a bit. Finally, should the loading control be GAPDH and not GADH?

13: In the results section, the authors refer to Fig. 9b instead of Fig. 9a when mentioning the differences in RNA-pol-II, CDK7/9 and BRD2/4 protein levels in 5-AZA sensitive versus resistant cells.

14: Fig. 9:

- Fig. 9a: Please indicate which of the lanes that contain OCI-M2 and SC, respectively, as well as M2AR1 and SCAR1, respectively.
- Fig. 9b and related text in the Results section: Nowhere is the information available which 5-AZA sensitive and corresponding resistant cell line that were used for this panel. Preferably should both OCI-M2/M2AR1 and SC/SCAR1 be investigated, but so far the authors only show one of these pairs and do not mention which.
- Fig. 9c-e: These experiments should preferably be performed also with SC vs SCAR1. Further: The corresponding figure legend for c-e could be shortened down to about one third of the current length...

15: What was the purity of the AML samples after Ficoll treatment, i.e. what was the percentage of leukemic cells versus normal mononuclear cells?

16: Fig. 10 and related text in the Results section:

- It is stated that 10 cases of each (MDS/RCMD, acute erythroid leukemia and acute monocytic leukemia) were analyzed, but data are only shown for one(?) case each. Are these representative for all cases analyzed?
- Did the MDS/RCMD case analyzed harbor any specific mutation of interest identified by exome sequencing?
- Please add information about magnification or similar in Fig. 10a (and in Supp. Fig. 4).
- It is a bit unclear what the grey bars in Fig. 10b actually show. Is it an average of positivity for

all 20 AML cases combined, or only showing either erythroid or monocytic leukemic cells, and in that case, which one? Also, how was the quantification performed?

- Fig. 10d: In the figure legend it is stated that hnRNPK and BRD4 are analyzed, but in the figure it says NSUN3. Which one is it?

- Legend is missing for Fig. 10e-f.

17: Why were only the MDS/RCMD samples subjected to exome sequencing, and not the primary AMLs as well? Further, the data should be available on a case-by-case basis as well and not only as the summary in Supp. Table 2, to allow the reader to look for co-occurrence of various mutations.

18: A large number of antibodies and other reagents are lacking in Supp. Table 3 (e.g. antibodies targeting hnRNPK, hnRNPA1, GAPDH, BRD2/4, etc.; sequences of the siRNAs targeting hnRNPK, DNMT2, NSUN3 as well as the control siRNA; several drugs, and so on). Please thoroughly go through the entire manuscript and add any missing reagents. Might also be preferred to split up the table in several tables as appropriate.

19: Please replace information regarding rpm in the ChIP protocol with rcf, since the actual speed might differ depending on which rotor type that is used...

20: Has "ml" been stated instead of "μl" in e.g. Step 1.4 ("...2μl glycogen, 120 ml phenol:chloroform...") in the CDMIA protocol? Please thoroughly check the entire protocol to make sure the correct unit has been used.

Reviewer #2 (Remarks to the Author):

The major claims of the manuscript are that hnRNPK directly binds RCTMs to form complexes relevant to the survival of leukemia cells, 5AZA causes disruption of active chromatin structure, leading to inhibition of leukaemia cells, and 5AZA resistant leukaemia cells have increased expression of RCTM and BRD4 bound RNA polymerase II, rendering these cells hypersensitive to BRD4 inhibitor. This is one of the first studies to demonstrate a positive correlation between 5Aza resistance and RCMT expression and a direct interaction between hnRNPA and RCMTs. Although the proposed mechanism is interesting and the potential clinical implications are significant, the findings are not sufficiently supported in appropriate model systems, and the major claims are overstated. Moreover, there are several concepts within the manuscript, but they appear disjointed. Below are questions, comments, and recommendations:

- Majority of the claims are supported by correlative data. Appropriate "rescue" experiments to establish a real functional mechanism in leukemia are needed.

- The leukemia cell line data is limited, which weakens the relevance of the study as it relates to leukemia pathogenesis and drug resistance. Not enough cell lines and patient samples were evaluated to convincingly establish that a particular phenotype is associated with a specific perturbation (i.e., 5Aza sensitivity, 5mC RNA levels, sensitivity to knockdown of hnRNPK/DNMT2/NSUN3, etc). It is not acceptable to compare only 1 cell line per group. Are 5mC RNA levels different in 5-Aza resistant and sensitive AML patient blasts?

- In Figure 2, is the binding of hnRNPK and RCTMs affected by 5-Aza?

- The siRNA data is not rigorous enough. siRNAs only produce a transient effect in leukemic cells. What was the efficiency of transfection? How many siRNAs were examined? It is suggested to establish knockdown with shRNAs or gRNA approaches, and perform rescue experiments.

- What is the rationale to perform 3C at the SPI/PU.1 locus?

- In Figure 7B, the claim that the SPI1/PU1-TET2 interaction is RNA-dependent is not clearly supported by the data. The quality of the IP is poor.

- In Figure 10c, the STED confocal microscopic data showing colocalization of hnRNPK and NSUN3 in AML is not appreciable.

Reviewers' comments:

Reviewer #1 (Remarks to the Author):

My concerns with this study are outlined below.

General points

1. It would be much easier to comment on the manuscript if the authors included page and/or line numbering.

Response: We have added page and line numbers to the revised manuscript.

2. The authors should have proofread the manuscript again prior to submission. There are now multiple instances of typos, duplication of words (e.g. "to measure to measure"), mistakes in various figures, etc., which give an unprofessional impression.

Response: We have carefully reviewed the revised manuscript.

Other comments

1. Fig. 1a: "Indicate which sample (left or right) that corresponds to OCI-M2 and SC, respectively, as well as M2AR1 and SCAR1."

Response: We have added the names of the leukaemia cell lines to the Fig. 1a.

2. Data related to Fig. 1b-c: “Have the authors performed the corresponding assay for SC and SCAR1 as well? This should be done, and be included at least as a “Data not shown”, depending on how interesting the results are.”

Response: We have included the data from SC and SCAR leukaemia cell lines to the Fig. 1b,c.

3. Supp. Fig. 1b: “Please add name of the cell line(s) in the legend and/or figure instead of only “...erythroid leukaemia cells...”.”

Response: We have added the names of leukaemia cell lines, OCI-M2, M2AR, SC and SCAR to all the figures.

4. Supp. Fig. 2a: “The authors have stated “Hypermethylation (5-hmC)” and “Methylation (5-mC)” in the top of this panel. Is that correct or should it refer to hydroxymethylcytosin and methylcytosin?”

Response: Thank you. We have corrected the typos in Fig. 2a.

5. Supp. Fig. 2b, d and e: “Y axis legend is missing.”

Response: We have added the Y axis labels to the supplementary Fig. 2b, d and e.

6. Data related to Fig. 2:

“Not stated here or in the corresponding Results section which cell line was used for these experiments. “

Response: We have added the names of cell lines, OCI-M2 and SC, for the RNase digestion-coupled IP/co-IP and siRNA knockdown experiments, respectively, in the result section, lines 131-134. In the same section, we also stated “Knockdown experiments using the same sets of siRNAs were performed in OCI-M2 cells and showed similar, but less dramatic effects, than those in SC cells due to a much lower transfection efficiency in OCI-M2 (~20% in OCI-M2 cells vs. ~70% in SC cells).”

“How many different siRNAs were evaluated per mRNA? At least two different ones are preferred to make sure any changes in cellular behavior etc. are not due to off-target effects. Especially for NSUN1/2 for which no knockdown was seen, additional siRNAs should be tested.”

Response: We are very grateful for the reviewer’s sturdy comments on our siRNA knockdown experiments. We obtained additional sets of commercially available siRNAs, three sets of siRNAs per gene, and performed NSUN1/NSUN2 knockdown experiments. See the siRNA info below (also provided in the supplementary info). As shown in Fig. 2c, we were able to successfully knockdown NSUN1/2, which had effects on the integrity of these proteins, but not the hnRNPk, NSUN3 or DNMT2.

Table S3. The sequences and the sources of the siRNAs used in this study.

Gene name	Sequence (5' to 3')	Sources/ cat. No.
HNRNPK	A: rCrGrArUrGrArArArCrCrUrArUrGrArUrUrArUrGrGrUrGGT B: rCrCrArArCrArCrUrArUrArArArGrGrArArGrUrGrArCrUTT C: rArGrUrArCrUrArCrArArGrUrUrGrArGrUrArArUrGrGrUAT	OriGene/ SR302173
NSUN1	A: rCrUrGrGrArCrUrArGrUrGrGrUrGrUrArUrGrATT B: rGrUrGrUrGrArUrCrCrUrUrGrCrCrArArUrGrATT C: rGrGrUrArGrArCrUrArUrGrCrUrCrUrGrArArATT	Santa Cruz/SC- 75962
NSUN2	A: rGrGrUrGrUrArGrArArArUrArArCrArGrCrGrGrUrGrArAGA B: rArGrArUrGrUrUrArArGrArUrArCrUrGrUrUrGrArCrCrCAG C: rArGrArArUrGrArArCrGrGrCrUrUrCrArUrUrArUrCrUrCAG	OriGene/ SR310319
NSUN3	A: rCrUrArCrArGrArUrArGrArGrCrUrGrUrUrArATT B: rCrUrCrUrGrGrGrUrCrUrGrUrUrUrGrGrArArUTT C: rCrUrCrUrGrGrGrUrCrUrGrUrUrUrGrGrArArUTT	Santa Cruz/SC- 78202
DNMT2 (TRDMT1)	A: rGrGrUrUrGrArGrArArUrArUrCrUrArCrArArUrCrCrCTT B: rGrGrArArUrGrUrArGrCrArUrGrArCrGrUrUrArArGrArUTT C: rGrCrArArCrArUrArCrArCrUrCrUrCrArArUrGrArArCrUTT	OriGene/ SR301245
BRD4	A: rCrUrGrArArCrCrUrCrCrCrUrGrArUrUrArCrUTT B: rCrCrArArCrUrGrCrUrArCrArArGrUrArCrArATT C: rCrArGrCrUrArArGrUrCrUrArGrArUrArUrCrATT	Santa Cruz/SC- 141740

“What is “100%” in Fig. 2c, since the control only reaches ~75%? These results suggest that even the control siRNA affects proliferation of the cells, which is a bit worrisome.”

Response: The difference was due to the transfection procedure, i.e. serum starvation, not due to control siRNAs. We have normalized the data and re-made the new fig 2d.

7. “At the top of the fourth page in the Results section, the authors refer to Table 1. Should this be Supplementary Table 1?”

Response: We have corrected this to be Supplementary Table 1, i.e. Table 1S.

8. Fig. 3: “Why are the order of high versus low concentration the opposite in a) and b) (hnRNP amount) compared to c) and d) (lysate amount)? Consistency would be preferred.”

Response: We have made the changes as suggested by the reviewer. Please note that the previous Figs. 3a,b, have been merged and rearranged as Fig. 1d, and the previous Figs. 3c,d have been rearranged as Fig. 1e,f in the revised new figures.

9. Data related to Fig. 4: “It is a bit bold to say that erythroid and monocytic cells have different chromatin structures at SPI1/PU.1, etc. when only one cell line per cell type has been investigated. The differences could potentially be cell line specific and not cell type specific. Another erythroid and monocytic cell line should be investigated for the authors to be able to make that statement.”

Response: We agree, and have added two additional sets of 3C and ChIP data from the erythroid leukaemia cell line, K562, and the monocytic leukaemia cell line, THP1. (See the new Figs. 3e,f. We have also changed the word “lineage-specific” to “lineage-associated” in the text. See the Results Section, lines 138-141.

10. Supp. Fig. 3d: “Both OCI-M2 and SC are mentioned in the figure legend, but only one bar per condition is shown in the figure. Is that an average of both cell lines or are only data from one of the cell lines shown? Data for both cell lines should be included in the figure in separate bars.”

Response: We have added the data figure from SC cells. Please see the revised Supplementary Figure 3e.

11. The heading of Fig. 6 and corresponding heading in the Results section: “The authors only show (and mention) data generated using OCI-M2, and can thereby not include the “erythroid vs. monocytic leukaemia” in the headings. For this, at least one (or preferably more) monocytic cell line(s) should be investigated. Further, three of the lower panels in a) are identical to three of the left hand panels in b) – please avoid this kind of redundancy! “

Response: We have dropped the redundant figure and added the data figures from the monocytic leukaemia cell line SC as the reviewer suggested. Please note the previous Fig. 6 has been rearranged as Figs. 3a,b.

12. Fig. 7: “The RNase amounts stated in all of the related panels suggest that duplicates of the different concentrations (5, 5, 25, 25) have been investigated on the gel/WB, but the corresponding results do not quite fit with what would be expected in that case - for instance are there highly different results for TET2 in the two different lanes stated to show data after treatment with 25K units of RNase in the top panel of a). Further, the blot for the TET2 co-IP in b) show all but trustable duplicates. Should it be a titration of four different concentrations and not only duplicates of two different concentrations? If this is a misunderstanding of the figure, that figure and corresponding legend need to be clarified quite a bit. Finally, should the loading control be GAPDH and not GADH?”

Response: We appreciate the reviewer’s thoroughness and accuracy. We have corrected the labelling errors in the revised Fig. 4d,e,f. Please note that the previous Fig. 7 has now been reassigned as Fig. 4.

13: “In the results section, the authors refer to Fig. 9b instead of Fig. 9a when mentioning the differences in RNA-pol-II, CDK7/9 and BRD2/4 protein levels in 5-AZA sensitive versus resistant cells.”

Response: The previous Figs. 9a and 9b have been changed to Fig. 6a and 6d in the revised manuscript.

14: Fig. 9a: "Please indicate which of the lanes that contain OCI-M2 and SC, respectively, as well as M2AR1 and SCAR1, respectively". Fig. 9b and related text in the Results section: "Nowhere is the information available which 5-AZA sensitive and corresponding resistant cell line was used for this panel. Preferably both OCI-M2/M2AR1 and SC/SCAR1 should be investigated, but so far the authors only show one of these pairs and do not mention which." Fig. 9c-e: "These experiments should preferably be performed also with SC vs SCAR1. Further: The corresponding figure legend for c-e could be shortened down to about one third of the current length..."

Response: We have indicated the lanes that contain OCI-M2, M2AR, SC and SCAR in the revised Fig. 6a (the previous Fig. 9a.). We have provided data from SCAR in Figs 6c,d. Please note that we also provided additional data from the reverse IP/co-IP, EU-clicking chemistry, proximity ligation and rolling circle amplification (PL-RCA), synergy between 5-AZA and JQ1 as well as NSUN1 siRNA knockdown in both the erythroid leukaemia cell lines, OCI-M2 and M2AR, and the monocytic leukaemia lines, SC and SCAR, in the revised Fig. 7 and 8. These experiments demonstrated the presence of distinct RCMT-associated active chromatin structures in these leukaemia cell lines.

15: "What was the purity of the AML samples after Ficoll treatment, i.e. what was the percentage of leukemic cells versus normal mononuclear cells?"

Response: We have included the blast count, immunophenotypes and other information for those AML cases used in this study in the revised supplementary table 2, i.e. Table 2S. We have also adjusted the data according to the blast counts in those specimens. See the revised Figs 9 and 10.

16: Fig. 10 and related text in the Results section: "It is stated that 10 cases of each (MDS/RCMD, acute erythroid leukemia and acute monocytic leukemia) were analyzed, but data are only shown for one(?) case each. Are these representative for all cases analyzed?"

Response: We have re-made the figures to provide the data from ten normal controls, ten MDS cases and ten AML cases in the revised Fig. 10c (the previous Fig. 10b).

17: "Why were only the MDS/RCMD samples subjected to exome sequencing, and not the primary AMLs as well? Further, the data should be available on a case-by-case basis as well and not only as the summary in Supp. Table 2, to allow the reader to look for co-occurrence of various mutations. Did the MDS/RCMD case analyzed harbor any specific mutation of interest identified by exome sequencing? Please add information about magnification or similar in Fig. 10a (and in Supp. Fig. 4). It is a bit unclear what the grey bars in Fig. 10b actually show. Is it an average of positivity for all 20 AML cases combined, or only showing either erythroid or monocytic leukemic cells, and in that case, which one? Also, how was the quantification performed? Fig. 10d: In the figure legend it is stated that hnRNPK and BRD4 are analyzed, but in the figure it says NSUN3. Which one is it? Legend is missing for Fig. 10e-f."

Response: Regarding sequencing analysis, the reason we did sequencing in the MDS/RCMD group was that we initially tried to identify possible causative mutations for those cytogenetically normal MDS. Since this manuscript is focusing on RCMT-mediated drug responsive/resistant active chromatin structures in leukaemia, we have removed the previous supplementary table S2 of MDS/RCMD NGS data. Instead, we have added several sentences in the last paragraph of the Discussion Section and the Supplementary Fig. 4 to describe our recent sequencing data from the 5-AZA-sensitive and 5-resistant leukaemia cell lines.

Please note that we have also provided additional data of mass spectrometric analysis of mRNA:m⁵C in 5-AZA-sensitive and 5-AZA-resistant MDS/AML bone marrow cells in the revised Fig. 9 and the PL-RCA of clinical 5-AZA-sensitive and 5-AZA-resistant MDS/AML bone marrow specimens in the revised Fig. 10a,b.

18: “A large number of antibodies and other reagents are lacking in Supp. Table 3 (e.g. antibodies targeting hnRNPK, hnRNPA1, GAPDH, BRD2/4, etc.; sequences of the siRNAs targeting hnRNPK, DNMT2, NSUN3 as well as the control siRNA; several drugs, and so on). Please thoroughly go through the entire manuscript and add any missing reagents. Might also be preferred to split up the table in several tables as appropriate.”

Response: We have made a new table (See the Supplementary Information Table 5S) to list all reagents used in this study.

19: Please replace information regarding rpm in the ChIP protocol with rcf, since the actual speed might differ depending on which rotor type that is used...

Response: We have made these corrections in the Supplementary Information.

20: Has “ml” been stated instead of “μl” in e.g. Step 1.4 (“...2μl glycogen, 120 ml phenol:chloroform...”) in the CDMIA protocol? Please thoroughly check the entire protocol to make sure the correct unit has been used.

Answer: We have corrected these changes in the Supplementary Information. Thank you.

Reviewer #2 (Remarks to the Author):

“The major claims of the manuscript are that hnRNPK directly binds RCTMs to form complexes relevant to the survival of leukemia cells, 5AZA causes disruption of active chromatin structure, leading to inhibition of leukaemia cells, and 5AZA resistant leukaemia cells have increased expression of RCTM and BRD4 bound RNA polymerase II, rendering these cells hypersensitive to BRD4 inhibitor. This is one of the first studies to demonstrate a positive correlation between 5Aza resistance and RCMT expression and a direct interaction between hnRNPA and RCMTs. Although the proposed mechanism is interesting and the potential clinical implications are significant, the findings are not sufficiently supported in appropriate model systems, and the major claims are overstated. Moreover, there are several concepts within the manuscript, but they appear disjointed. Below are questions, comments, and recommendations:

Majority of the claims are supported by correlative data. Appropriate “rescue” experiments to establish a real functional mechanism in leukemia are needed.

Response: We have established the novel working model of RNA:m⁵C/RCMT-mediated 5-AZA responsive/resistant chromatin structures in leukaemia cells based our original experimental data. We used various newly developed technologies, including STED confocal microscopy, EU-clicking chemistry and PL-RCA, to visualize such chromatin structures. Furthermore, we used different means, including various siRNAs and chemical compounds/inhibitors, to test and confirm our working models. We also demonstrated significantly increased mRNA:m⁵C and presence of NSUN1/BRD4-associated active chromatin structure in 5-AZA-resistant clinical specimens.

Regarding the suggestion of appropriate rescue experiments, those “rescue” experiments are not feasible because there is little known about the physiological functions and pathologic role of these RCMTs in leukaemogenesis, and more importantly, our data demonstrate that the 5-AZA resistance-associated chromatin structure is mainly due to the changes in RCMT interactions, i.e. the interactions between NSUN1 and BRD4/RNA-pol-II CTD-S2P, but not expression levels of RCMTs. We cannot perform “rescue” or targeting experiments until we identify the causes for the aberrant RCMT interactions.

“The leukemia cell line data is limited, which weakens the relevance of the study as it relates to leukemia pathogenesis and drug resistance. Not enough cell lines and patient samples were evaluated to convincingly establish that a particular phenotype is associated with a specific perturbation (i.e., 5Aza sensitivity, 5mC RNA levels, sensitivity to knockdown of hnRNPK/DNMT2/NSUN3, etc). It is not acceptable to compare only 1 cell line per group. Are 5mC RNA levels different in 5-Aza resistant and sensitive AML patient blasts?”

Response: Regarding the leukaemia cell lines used in this study, as shown in the supplementary Table 1S, we have tested 10 myeloid leukaemia cell lines, which were all the myeloid leukaemia cell lines available to us. We focused on SC and OCI-M2 and their derived 5-AZA-resistant leukaemia cell lines because, 1) they are the leukaemia cells representative of monocytic and erythroid lineages and our initial data demonstrated distinct drug response patterns in monocytic vs. erythroid leukaemia cells, and 2) unlike other myeloid leukaemia cell lines, SC and OCI-M2 leukaemia cells do not contain recurrent chromosome translocations. This is important because a majority of clinical AML/MDS cases do not have recurrent chromosome translocations. Therefore, SC and OCI-M2 more truthfully reflect the clinical scenario.

Regarding measuring RNA:m⁵C levels in clinical leukaemia blasts, we have provided the data of mass spectrometric analysis of mRNA:m⁵C in the revised Fig. 9. One should realize that it is very difficult to perform such m⁵C analysis, even in our state-of-the-art RNA methylation lab, because m⁵C only accounts for 1-2% of total RNA modifications. It is very difficult to have sufficient amount of pure mRNA from clinical leukaemia specimens for m⁵C analysis. Another difficulty is that the leukaemia blasts in a large portion of those clinical specimens are negative for CD34, a marker often used for selecting leukaemia blasts. To our knowledge, our mass spectrometric analysis is one of the first attempts to determine mRNA: m⁵C in clinical specimens.

“In Figure 2, is the binding of hnRNPK and RCTMs affected by 5-Aza?”

Response: This question was addressed by the previous Fig. 8d. Our previous Fig. 8d clearly demonstrated that the interactions between hnRNPK and RCMTs were not affected by 5-AZA. The previous Fig. 8d has been rearranged as Fig. 5e in the revised figures. The following is the previously submitted Fig. 8d.

“The siRNA data is not rigorous enough. siRNAs only produce a transient effect in leukemic cells. What was the efficiency of transfection? How many siRNAs were examined? It is suggested to establish knockdown with shRNAs or gRNA approaches, and perform rescue experiments.”

Response: We have provided answers to the above questions as we addressed the 1st reviewer’s question 6. We have also tried to explain why “rescue” experiments are not feasible for this case. However, we indeed tried to use synergy experiments (5-AZA plus NSUN1 siRNAs and 5-AZA plus JQ) to provide the answers of so-called “recuse” experiments in this case.

“What is the rationale to perform 3C at the SPI/PU.1 locus?”

Response: Because SPI1/PU.1 is a key regulator of myelopoiesis. We and others have demonstrated marked changes in DNA/histone modifications, chromatin conformation and chromatin modifier in the SPI1/PU.1 regulatory regions in clinical specimens of hematologic neoplasms.

“In Figure 7B, the claim that the SPI1/PU1-TET2 interaction is RNA-dependent is not clearly supported by the data. The quality of the IP is poor.”

Answer: The SPI1/PU.1-TET2 interaction is NOT RNA-dependent, i.e. RNase-resistant. Our revised Fig. 4f,g (previously Fig. 7b) specifically addresses the RNA-dependence of those interactions. See the figures below.

Figure 4

We have been working on TET2 for several years. IP and co-IP with TET2 are very difficult because this protein is very large and easily degraded.

“In Figure 10c, the STED confocal microscopic data showing colocalization of hnRNPK and NSUN3 in AML is not appreciable.”

Response: It is a partial co-localization of hnRNPK and NSUN3 because NSUN3 is diffusely expressed in the cytoplasm and nuclei of the marrow cells. AML shows more dramatic increase in the co-localization of hnRNPK and RNA-pol-II because hnRNPK is more localized in the nuclei of AML leukaemia cells. However, to avoid any confusion, we have removed the previous figure 10c,d of STED confocal analysis of hnRNPK/NSUN3 co-localization in clinical specimen in the revised manuscript.

Reviewers' comments:

Reviewer #1 (Remarks to the Author):

The authors have responded to my concerns in a satisfactory way, and I only have a minor comment for their potential consideration:

Perhaps not quite optimal to introduce new data (related to Fig 4S) in the Discussion section instead of in the Results section.

Reviewer #2 (Remarks to the Author):

The authors made a significant effort to address the original concerns. The new data and revised manuscript are clearer and supported by additional supporting experiments. In particular, the new MDS/AML patient data advances the major claims of the paper. Although the manuscript is dense and addressing a complicated mechanism, the findings are interesting and the potential clinical implications are significant. I appreciate the concerns that "rescue" experiments are not technically straightforward; however, this still remains a lingering issue in order to establish unequivocal mechanistic links and causal phenotypes.

- As indicated above, the manuscript is dense. Certain figures can go into the supplemental file.
- Why does knockdown of hnRNPK result in reduced steady-state expression of DNMT2, NSUN3, and PU.1? Can restored DNMT2 and NSUN3 expression rescue the hnRNPK-deficient phenotype (such as in Figure 2d)?
- The transfection efficiency of the siRNA is not adequately addressed. What is the level of knockdown at day 3 (time point used for the phenotypic studies)?
- Please provide a reference or show the relevant today to support this following statement: We screened a pool of RNA-binding proteins (RBPs) using antibodies against GATA1 (in OCI.M2) and SPI1/PU.1 (in SC) as IP "baits" to identify the interacting RNP(s).

We wish to thank the reviewers for reviewing our manuscript and providing us their comments that will strengthen our paper.

Please see our pointa bya point responses to the reviewers' comments below. We have highlighted all changes in yellow in the revised manuscript as instructed.

Response to review:

Reviewer #1: "The authors have responded to my concerns in a satisfactory way, and I only have a minor comment for their potential consideration: Perhaps not quite optimal to introduce new data (related to Fig 4S) in the Discussion section instead of in the Results section."

We agree with the reviewer. We have moved the NGS figures from the Discussion to the Results (lines 378a 390).

Reviewer #2: "The authors made a significant effort to address the original concerns. The new data and revised manuscript are clearer and supported by additional supporting experiments. In particular, the new MDS/AML patient data advances the major claims of the paper. Although the manuscript is dense and addressing a complicated mechanism, the findings are interesting and the potential clinical implications are significant. I appreciate the concerns that "rescue" experiments are not technically straightforwardI however, this still remains a lingering issue in order to establish unequivocal mechanistic links and causal phenotypes."

h "The manuscript is dense. Certain figures can go into the supplemental file."

We appreciate the reviewer's suggestion. We have moved as many figures as possible into the supplemental file.

a "Why does knockdown of hnRNPK result in reduced steadyh state expression of DNMT2, NSUN3, and PU.1?"

The reason for reduction of these proteins by knockdown of hnRNPK is that hnRNPK binds these proteins directly (Fig. 2) to form a functional complex that is important for maintaining the integrity of the associated proteins such as DNMT2 and NSUN3. Epigenetic/chromatin

modifying complexes, SAGA or SWI/SNF complexes for example, are composed of multiple components, and deletion of the key components in the complexes results in degradation of other associated proteins and destruction of the entire complex (Hassan AH, Neely KE, Workman JL. *Cell*. 2001 104(6):817-27; Marmorstein R, Trievel RC. *Biochim. Biophys. Acta*. 2009, 1789: 58–68.

-“Can restored DNMT2 and NSUN3 expression rescue the hnRNPK-deficient phenotype (such as in Figure 2d)?”

We appreciate the reviewer’s suggestion of rescuing experiments with expression of DNMT2 and NSUN3. Unfortunately, to our knowledge, such rescuing experiments are technically infeasible. DNMT2 has many isoforms, identified as “a” to “k”. Little is known about the *in vivo* functions and expression levels of these isoforms. The cDNAs (mRNAs) of these isoforms are huge (>13,000 bp for each isoform), and they are not commercially available. Therefore, it is technically impossible to do rescuing experiments when dealing with so many unknown isoforms and such large molecules. Also, enforcing expression of such large proteins *in vivo* is often toxic to cells and unlikely to yield a meaningful outcome.

- “The transfection efficiency of the siRNA is not adequately addressed.”

We appreciate the reviewer’s comment. We have added the following paragraph to describe the method and efficiency of the siRNA transfection technology that we used in the Experimental Procedures.

***In Vivo* Transfection Assay**

Lipofectamine™ RNAiMAX Reagent (Cat. 13778075, Invitrogen) was used to transfect leukaemia cells with siRNAs targeting hnRNPK and various RCMTs. The transfection assays were performed by following the company-provided forward transfection protocol for bone marrow cells. The KDaIert™ GAPDH Assay Kit (Cat. AM1639, Invitrogen) was used to optimize transfection conditions and to assess for transfection efficiency by following the company-provided protocol.

We have also added the following sentence in the Results Section (line 133-135).

The transfection efficiencies for OCI-M2 cells and SC cells were ~20% and ~70%, respectively, measured by using KDaIert™ GAPDH Assay Kit.

- “What is the level of knockdown at day 3 (time point used for the phenotypic studies)?”

The siRNA knockdown experiments in the figure 2b and 2c were performed at day 3 as stated in the corresponding figure legend, and the western blotting clearly demonstrates the levels of knockdown of RCMTs by siRNAs. See below.

- *“Please provide a reference or show the relevant data to support this following statement: We screened a pool of RNA.binding proteins (RBPs) using antibodies against GATA1 (in OCI.M2) and SPI1/PU.1 (in SC) as IP “baits” to identify the interacting RNP(s).”*

We have added a paragraph with the related references in the Discussion section (lines 397-407). We also provide the relevant data in Fig. 4f and the supplementary Fig. 6S.

REVIEWERS' COMMENTS:

Reviewer #2 (Remarks to the Author):

I have no further comments or concerns.